# Very long-chain fatty acids drive 1-deoxySphingolipid toxicity

Adam Majcher[1], Gergely Karsai [1], Elkhan Yusifov [1], Martina Schaettin[2], Ermanno Malagola[1], Peter Horvath[1], Jinmei Li[3], Sofía Rodriguez-Gallardo [4], Kuniyoshi Shimizu [4,5], Gai Zhibo[6], Raghvendra Dubey[2], Tim Peterson[3], Takeshi Harayama [4] & Thorsten Hornemann [1] ✉

1-Deoxysphingolipids (1-deoxySLs) are atypical sphingolipids formed when serine palmitoyltransferase incorporates L-alanine instead of L-serine. Elevated 1-deoxySLs are associated with hereditary sensory neuropathy type 1 and diabetic neuropathy, but the molecular basis of their toxicity remains unclear. Here we show that toxicity is mediated by very long-chain (VLC) 1-deoxy-dihydroceramides (1-deoxyDHCer), particularly nervonyl-1-deoxyDHCer (m18:0/24:1) and lignoceryl-1-deoxyDHCer (m18:0/24:0). Using a CRISPR interference screen, we identify ELOVL1 and CERS2 as essential enzymes driving the formation of these toxic species. Genetic modulation or pharmacological inhibition of ELOVL1 prevents VLC 1-deoxyDHCer accumulation, rescuing the toxicity in cellular and neuronal models. Mechanistic studies reveal that m18:0/24:1 disrupts mitochondrial integrity and induces the mitochondrial permeability transition pore formation and BAX activation, leading to cell death. These findings establish a direct link between 1-deoxySL chemical structure and cytotoxicity and highlight ELOVL1 inhibition as a potential therapeutic strategy for 1-deoxySL-associated diseases.

Sphingolipids (SL) are a structurally diverse group of bioactive lipids that share a long-chain base (LCB) as a common core structure. Forming the LCB is the rate-limiting reaction in the SL de novo synthesis and initiates with the conjugation of L-Serine and Palmitoyl-CoA, catalyzed by the Serine-Palmitoyltransferase (SPT). The resulting product 3-ketosphinganine is rapidly converted by ketodihydro-sphingosine reductase (KDSR) to Sphinganine (Sa, d18:0), which serves as a substrate for a group of Ceramide synthases (CERS) that N-acylate the LCB with a long chain (LC) - or very long chain (VLC) fatty acid. Mammals express six CERS isoforms (CERS1-6) with different specificities towards the length and saturation state of their fatty acid substrates. The product, dihydroCeramide (DHCer), is finally converted to Ceramides (Cer) by the Dihydroceramide desaturase 1 (DEGS1), by introducing a Δ4E double bond into the LCB backbone. Ceramides are the central molecules of SL metabolism and serve as building blocks for complex SL, such as Sphingomyelins and Glycosphingolipids[1].

In addition to L-Serine, SPT can also utilize L-Alanine, forming an atypical LCB 1-deoxysphinganine (1-deoxySa, m18:0) which, in comparison to Sa, lacks the essential hydroxyl group at C1 position[2]. Like Sa, also 1-deoxySa is metabolized by CERS, forming 1-deoxy-dihydroCeramides (1-deoxyDHCer, m18:0/xx:y)[3], but in contrast to canonical SL, 1-deoxyDHCer are not further metabolized by DEGS1 but instead by fatty acyl desaturase 3 (FADS3), which introduces a Δ14Z, instead of a Δ4E double bond[4].

As 1-deoxySL lacks the C1-hydroxylic group of canonical LCBs, they cannot be converted into complex SL, but also not degraded via the canonical SL catabolism, which requires the formation of the catabolic intermediate Sphingosine-1-Phosphate (S1P)[3].

[1]Institute of Clinical Chemistry, University Hospital Zurich, Zurich, Switzerland. [2]Department of Reproductive Endocrinology, University Hospital Zurich, Schlieren, Switzerland. [3]Bioio Inc., St. Louis, MO, USA. [4]Institut de Pharmacologie Moléculaire et Cellulaire, Université Côte d'Azur - CNRS UMR7275 - Inserm U1323, Valbonne, France. [5]Department of Biotechnology and Life Science, Tokyo University of Agriculture and Technology, 184-8588 Tokyo, Japan. [6]Department of Clinical Pharmacology and Toxicology, University Hospital Zurich, Zurich, Switzerland. ✉e-mail: thorsten.hornemann@usz.ch

1-deoxySL are toxic to neurons and other cells[4–8], and have been linked to peripheral sensory neuropathies such as the Hereditary Sensory Neuropathy Type 1 (HSAN1)[9–12], diabetic neuropathy[13–16], as well as the rare retinopathy Macular Telangiectasia type 1 (MacTel)[17]. 1-deoxySLs have been shown to trigger various pathological cellular processes, such as altered mitochondrial fission, increased radical oxygen species (ROS), autophagy, endoplasmic reticulum (ER) stress, unfolded protein response, cytoskeletal defects, and calcium handling abnormalities[5,6,8,18–21]. Despite these findings, the toxicity of 1-deoxySL and the underlying molecular mechanisms are not yet understood.

In this work, we combine a CRISPRi-based functional genomics screen with functional lipidomics, targeted genetic interference, and pharmacological studies to understand the metabolic and structural determinants of 1-deoxySL toxicity.

## Results

### 1-deoxyDHCeramides are the primary mediators of 1-deoxySphingolipids induced toxicity

To investigate the molecular mediators of 1-deoxysphingolipid (1-deoxySL) toxicity, we employed the ceramide synthase (CERS) inhibitor Fumonisin B1 (FB1), a mycotoxin previously reported to mitigate 1-deoxySa-induced cytotoxicity (Fig. 1 a)[20,22].

Co-treatment of HeLa cells with 1-deoxySa and FB1 led to a substantial reduction in toxicity, confirming that CERS inhibition has a protective effect (Fig. 1b). To investigate the underlying metabolic changes, we conducted a stable isotope-labeled flux assay by supplementing d$_3$-1-deoxySa for 24 hours, followed by high-resolution LC-MS/MS-based lipidomics to trace the incorporation of the labeled LCB into downstream sphingolipid species. FB1 treatment led to a marked accumulation of free long-chain bases (LCBs), including both d$_3$-1-deoxySa and d$_3$-1-deoxySo, while simultaneously suppressing the formation of N-acylated products, including d$_3$-1-deoxyDHCer and d$_3$-1-deoxyCer (Fig. 1c, d). These findings suggest that toxicity is not mediated by the LCBs themselves, but rather by their N-acylated derivatives.

Next, we investigated whether the presence of the Δ14Z double bond has an influence on toxicity. To assess the role of the Δ14(Z) double bond introduced by fatty acid desaturase 3 (FADS3, Fig. 1a), we compared the cytotoxicity of 1-deoxySa (saturated, m18:0) and its unsaturated analog 1-deoxySo (m18:1, Δ14(Z)). Consistent with previous reports implicating FADS3 in detoxification[4], the addition of 1-deoxySa exhibited notably higher cytotoxicity compared to 1-deoxySo (Fig. 1e). 1-deoxySo did not achieve dose-dependent 50% inhibition within the tested range (0–3 μM), and an IC$_{50}$ could not be determined. To determine the metabolic fate of these LCBs, we performed an untargeted lipidomics of HeLa cells supplemented with either 1-deoxySa (m18:0) or unsaturated 1-deoxySo (m18:1, Δ14(Z)). Supplementation with 1-deoxySa resulted in the accumulation of both 1-deoxyDHCer and 1-deoxyCer species, whereas 1-deoxySo was metabolized exclusively to 1-deoxyCer (Fig. 1f,g). Together, these data indicate that saturated N-acylated species, particularly 1-deoxyDHCer, are the principal mediators of 1-deoxySL-induced cytotoxicity.

### CRISPRi toxicity screen identifies gene candidates responsible for 1-deoxySphingolipids toxicity

To identify genes involved in 1-deoxySL toxicity, we performed a systematic CRISPRi toxicity screen. K562 cells expressing CRISPRi-dCas9-KRAB were transfected with a CRISPRi v2 sgRNA library and cultured for five passages in the presence of 1-deoxySa (1.5 μM) which corresponds to the experimentally determined IC$_{50}$ under the screening conditions (Fig. 2a). Comparative analysis of sgRNA abundance revealed 5 genes whose knockdown conferred a significant resistance to 1-deoxySa induced toxicity (Fig. 2b). All identified genes are functionally linked to the elongation of fatty acids (ACACA, HSD17B12, PTPLB), the synthesis of VLC-FAs (ELOVL1) or the incorporation of

VLC-FAs into the LCB backbones (CERS2) (Fig. 2c). Notably, ELOVL1 catalyzes the elongation of saturated and monounsaturated acyl-CoAs to VLC-FAs such as 22:0, 24:0, and 24:1, which are the substrates for CERS2, forming VLC-1-deoxyDHCer species[23]. These findings position ELOVL1 directly upstream of CERS2 in the biosynthetic VLC-FA pathway. Conversely, FADS3 was identified as a sensitizing gene, consistent with the protective effect of the Δ14(Z) double bond observed in earlier experiments (Fig. 1e). To further assess the functional context of these hits, we performed gene set enrichment analysis (GSEA) using logFC values derived from the CRISPRi screen. This revealed significant enrichment of genes involved in the biosynthesis of unsaturated fatty acids and elongation among those promoting toxicity (Fig. 2d). In contrast, protective gene signatures were more broadly distributed across pathways related to ribosomal function, oxidative phosphorylation, DNA replication, RNA transport, and peroxisomal activity (Fig. 2d). This highlights the central role of the lipid metabolic networks, particularly VLC-FA biosynthesis, in mediating 1-deoxySL-induced cytotoxicity.

### Silencing of ELOVL1 and CERS2 expression confirms their role in 1-deoxysphingolipid-mediated toxicity

Based on the CRISPRi screen, we focused on the functional validation of ELOVL1 and CERS2 as key mediators of 1-deoxySL toxicity (Fig. 3a).

To assess the role of ELOVL1, we performed siRNA-mediated knockdown in HeLa cells. RT-qPCR confirmed selective silencing of ELOVL1, with no off-target effects on other ELOVL isoforms (Fig. 3b). To evaluate the metabolic consequences, we conducted a d$_3$-1-deoxySa isotopic flux assay coupled to high-resolution LC-MS/MS lipidomics. ELOVL1 knockdown led to a significant reduction in very-long-chain (VLC) 1-deoxyDHCer species, specifically in m18:0/24:0 and m18:0/24:1, and a compensatory increase in long-chain (LC) 1-deoxyDHCer species (m18:0/16:0, m18:0/18:0, m18:0/20:0) (Fig. 3c). This shift in sphingolipid composition was accompanied by a substantial reduction in 1-deoxySa-induced cytotoxicity (Fig. 3d).

To further confirm these findings, we used CRISPR-Cas9-generated ELOVL1 knockout (KO) HeLa cells. As expected, ELOVL1 KO resulted in decreased flux of d$_3$-1-deoxySa into VLC 1-deoxyDHCer species, with a corresponding increase in LC 1-deoxyDHCer species, recapitulating the knockdown phenotype (Fig. 3e). These data establish ELOVL1 as a critical upstream enzyme modulating the accumulation of cytotoxic VLC 1-deoxyDHCer species.

We next examined the roles of the CERS2 and CERS5/6 isoforms. While CERS2 primarily utilizes VLC acyl-CoAs (22:0, 24:0, 24:1), CERS5 and CERS6 preferentially process LC fatty acyl-CoAs (16:0, 18:0). To investigate their functions, we used CRISPR-Cas9 to generate CERS2 knockout (KO) and CERS5/6 double KO HeLa cell lines, which were compared to empty vector (EV) controls.

CERS2 KO cells showed a significant reduction in d$_3$-1-deoxySa-derived VLC 1-deoxyDHCer species (m18:0/22:0, m18:0/24:0, m18:0/24:1), along with elevated levels of LC 1-deoxyDHCer and free LCB 1-deoxySa (Fig. 3g). In contrast, CERS5/6 KO cells had a reduced formation of LC 1-deoxyDHCer species, with minimal effects on VLC species. Functional assays confirmed that the absence of CERS2 conferred substantial protection against 1-deoxySa toxicity, whereas CERS5/6 KO cells had only a marginal influence on the 1-deoxySL-induced toxicity (Fig. 3h).

Together, these results validate ELOVL1 and CERS2 as key enzymes driving the synthesis of cytotoxic VLC 1-deoxyDHCer species and mediating 1-deoxySL-induced cell death.

### Pharmacological inhibition of ELOVL1 mitigates 1-deoxysphingolipid-induced mitochondrial and neuronal toxicity

To evaluate whether a pharmacological inhibition of ELOVL1 could serve as a potential therapeutic strategy to reduce 1-deoxySL toxicity,

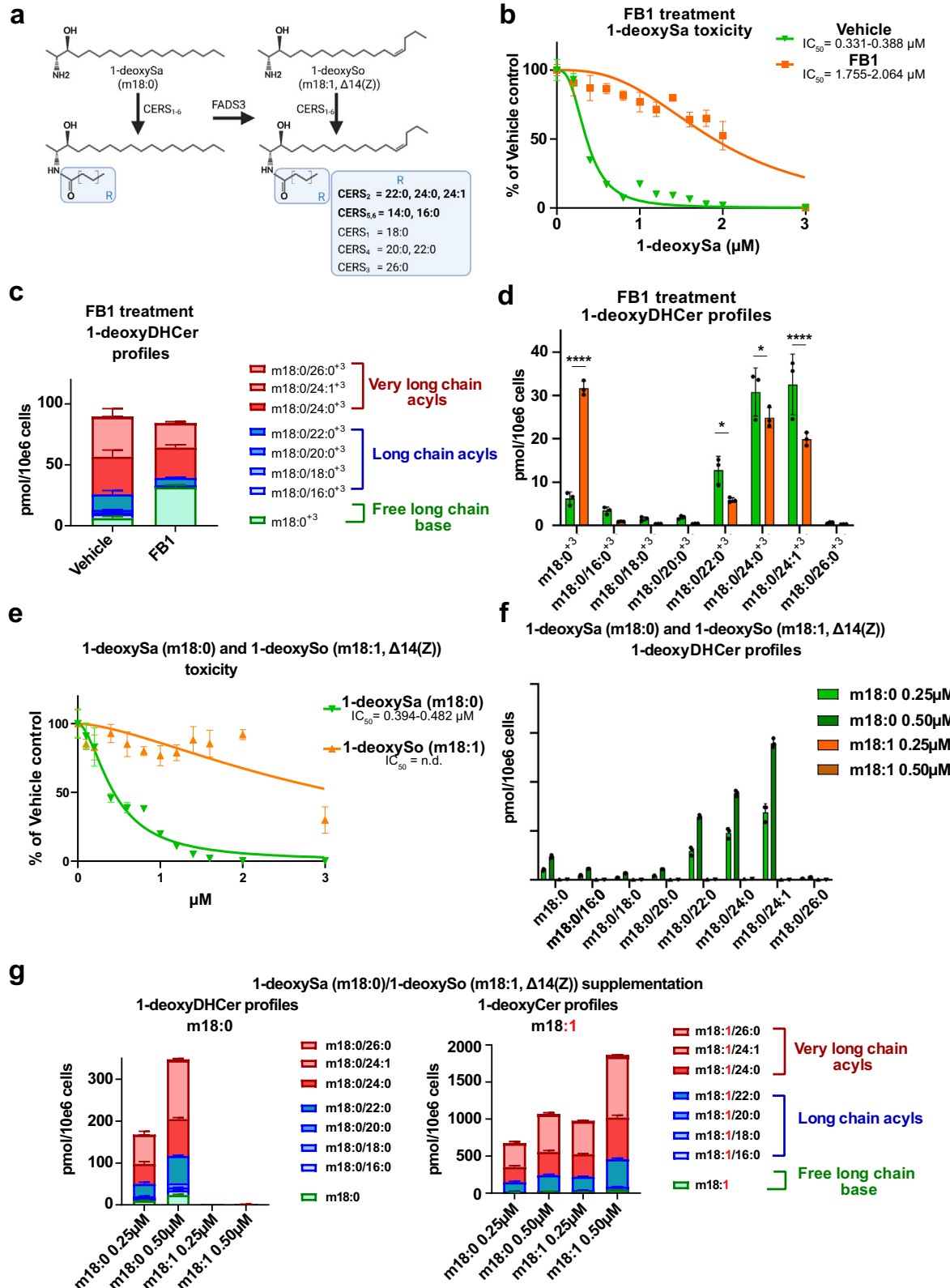

we employed the previously reported small-molecule inhibitor Compound 22 (22) (CAS 2761063-99-2). The pyrimidine-ether compound has been shown to be biologically active in mouse and rat models, with established repeat-dose toxicology studies in cynomolgus monkeys[24,25]. This compound was originally developed as a potent and selective ELOVL1 inhibitor in the context of X-linked

adrenoleukodystrophy (X-ALD) with well-characterized in vitro and in vivo activities (Fig. 4a)[24,25].

In HeLa cells, treatment with (22) significantly reduced the metabolic flux of $d_3$-1-deoxySa into very-long-chain (VLC) 1-deoxyDHCer species (m18:0/24:0 and m18:0/24:1), and again caused a compensatory increase in LC-1-deoxyDHCer species (m18:0/16:0,

**Fig. 1 | 1-deoxyDHCeramides are the primary mediators of 1-deoxySphingolipids induced toxicity. a** Schematic representation of 1-deoxySL metabolism. The horizontal arrow indicates desaturation of 1-deoxysphinganine (1-deoxySa, m18:0) at the Δ14(Z) position catalyzed by fatty acid desaturase 3 (*FADS3*). Vertical arrows represent N-acylation of 1-deoxySa by ceramide synthase isoforms (*CERS1–6*). The blue box highlights the acyl-CoA substrate specificities of individual *CERS* isoforms. **b** 1-deoxySa (m18:0) dose-response toxicity curves in HeLa cells in the presence or absence of the pan-CERS inhibitor fumonisin B1 (FB1, 35 μM). Toxicity was determined by CellTiter-GLO assay as described in the methods section. $IC_{50}$ values are indicated below the individual treatments. **c, d** Formation of $d_3$-1-deoxyDHCer (m18:0/xx:y[+3]) species in HeLa cells supplemented with $d_3$-1-deoxySa and treated with pan-Ceramide synthase inhibitor Fuminosin B1 (FB1, 35 μM) or Vehicle. **e** Comparison of cytotoxicity of 1-deoxySa or (m18:0) 1-deoxysphingosine (1-deoxySo, m18:1, Δ14(Z)) on HeLa cells. **f, g** Intracellular accumulation of 1-deoxyDHCer and 1-deoxyCer species formed in HeLa cells supplemented with 1-deoxySa (m18:0) or 1-deoxySo (m18:1, Δ14(Z)). All lipid data experiments were performed with $n = 3$ biologically independent replicates (separately cultured and treated HeLa cell populations) and measured by an untargeted high-resolution LC-MS/MS lipidomics workflow and normalized to an internal standard and cell count. Data are shown as mean ± SD. Significance was determined using a two-tailed unpaired Student's t-test with multiple testing correction (two-stage step-up method of Benjamini, Krieger, and Yekutieli). Reported p.adjusted values: $p < 0.05$ (*), $p < 0.01$ (**), $p < 0.001$(***). Exact *p*-values and statistical source data are provided in Source Data 1. Toxicity data were normalized to vehicle treated cells and represented as mean ± SD ($n = 4$, separately cultured and treated HeLa cell populations). Non-linear regression analysis was used to calculate toxicity curves and $IC_{50}$ values (95% Confidence Interval). Illustrations were created in BioRender. Hornemann, T. (2025) https://BioRender.com/ifocfkk.

m18:0/18:0, m18:0/20:0) (Fig. 4b), resulting in a significant protection from 1-deoxySa-induced cytotoxicity (Fig. 4c).

Previous studies have linked elevated levels of 1-deoxySLs to mitochondrial dysfunction and structural damage, including mitochondrial fragmentation[19]. We therefore tested whether compound 22 could prevent 1-deoxySL-induced mitochondrial damage. We performed a Seahorse XF Mito Tox assay in HeLa cells, treated with 1-deoxySa (0.5 μM). We observed a marked reduction in mitochondrial respiration (OCR), glycolytic capacity (ECAR), and maximal respiratory function. A co-treatment with (22) fully restored all functional parameters (Fig. 4d–f). All mitochondrial data were normalized to crystal violet staining to account for cell loss after 1-deoxySa treatment and enable accurate assessment of mitochondrial activity.

To confirm the metabolic and protective effects of (22) in a physiological neuronal context, we established an in vivo model based on a live chicken embryo model. Living chicken embryos were injected with $d_3$-1-deoxySa and treated with or without (22) as described in the methods section, followed by dissection of dorsal root ganglia (DRG) and high-resolution LC-MS/MS lipidomics profiling (supplementary Fig. 1a). Treatment with (22) significantly reduced the formation of $d_3$-labelled 1-deoxyDHCer (m18:0/24:1). Additionally, treatment with (22) significantly reduced the formation of VLC 1-deoxyDHCer and VLC 1-deoxyCer species (m18:0/22:0, m18:0/23:0, m18:0/24:1, m18:1/22:0, m18:1/23:0, and m18:1/24:1) with compensatory increase in LC 1-deoxyDHCer and LC 1-deoxyCer (m18:0/:18:0, m18:0/20:0, m18:1/:18:0 and m18:0/20:0) (supplementary Fig. 1b–d). These results indicate that (22) is metabolically active in the nervous system, particularly in sensory neurons, which are the primary targets of 1-deoxySL toxicity. However, due to technical limitations, the effects on the developing peripheral nervous system of the embryo have not yet been evaluated. Instead, we assessed the neuroprotective potential of (22) in dissociated chicken DRG neurons obtained from day-9 chicken embryos by supplementing 1-deoxySa (0.05 or 0.1 μM) in the presence or absence of (22). Neurons were stained for neurofilament-M, and neurite area was quantified by fluorescence microscopy and image analysis (supplementary Fig. 1e). 1-deoxySa treatment caused a significant loss of neurites, which was effectively rescued by (22) (supplementary Fig. 1f, g). Notably, (22) alone had no adverse effect on neurite morphology, indicating that ELOVL1 inhibition itself is not intrinsically neurotoxic.

In summary, these findings demonstrate that compound (22), a preclinically validated pharmacological ELOVL1 inhibitor, protects against 1-deoxySL-induced mitochondrial dysfunction and neurotoxicity, supporting its potential as a therapeutic strategy for disorders involving pathological 1-deoxySL accumulation.

## Nervonyl-1-deoxyDHCeramide (m18:0/24:1) is the strongest inductor of mitochondrial damage and cellular toxicity

Finally, we aimed to see whether toxicity is different for the individual LC and VLC 1-deoxyDHCer species (Fig. 5 a). ELOVL1 inhibition prevents the formation of VLC 1-deoxyDHCer synthesis but does not directly inhibit CERS2 activity. Therefore, we aimed to resupplement individual fatty acid substrates (16:0, 24:0, and 24:1) in the presence of (22) to address the toxicity of the individual 1-deoxyDHCer species. Co-treatment with (22) was essential to prevent elongation of supplemented fatty acids, such as palmitic acid (16:0), which would otherwise be metabolized by ELOVL1 into 24:0 and 24:1 fatty acids. Additionally, as VLC-FA's are typically not absorbed efficiently by the cells, we developed a protocol that allowed an efficient resorption of the supplemented FAs. For that, FAs were supplemented as Na+ salts in complex with bovine serum albumin (BSA) (methods section) and added in the presence of (22) to block further elongation.

Lipidomics analysis demonstrated that fatty acid supplementation selectively elevated the levels of matching LC or VLC 1-deoxyDHCer species (Fig. 5b). Toxicity assays revealed that primarily nervonate (24:1), which promoted the formation of 1-deoxyDHCer (m18:0/24:1) (also referred as nervonyl-1-deoxyDHCer), induced the strongest cytotoxic effect, followed by lignocerate (24:0) and palmitate (16:0) (Fig. 5c). These findings identify nervonyl-1-deoxyDHCer (m18:0/24:1) as the most toxic 1-deoxySL species generated in the cells.

To evaluate the impact of these individual species on mitochondrial integrity, we conducted a mitochondrial swelling assay using freshly isolated mitochondria exposed directl7y to synthetic LC and VLC 1-deoxySL. Among the tested species, 1-deoxyDHCer (m18:0/24:1) induced the most signficant swelling reponse, followed by 1-deoxyDHCer (m18:0/24:0) and (m18:0/16:0), whereas unsaturated 1-deoxyCer (e.g., m18:1(4E)/16:0, m18:1(4E)/24:0) and canonical dihydroceramides (d18:0/xx:y) showed no effect (Fig. 5d). It is important to note that due to commercial unavailability of physiologically relevant 14Z 1-deoxyCer, 4-5E isomers were used. Anyway, this demonstrates that 1-deoxyDHCer (m18:0/24:1) and 1-deoxyDHCer (m18:0/24:0) directly perturb mitochondrial integrity.

Mitochondrial swelling is a hallmark of mitochondrial permeability transition pore (mPTP) opening, a well-known trigger for apoptosis[26]. To test whether mPTP is involved in 1-deoxySL-induced toxicity, we treated isolated mitochondria with 1-deoxyDHCer (m18:0/24:1) in the presence or absence of the mPTP inhibitor cyclosporin A (CsA). CsA decreased the mitochondrial swelling induced by this species (Fig. 5e). Furthermore, in cellular toxicity assays, CsA rescued HeLa cells from 1-deoxySa-induced death (Fig. 5f), implicating mPTP activation in the cytotoxic mechanism.

Since mPTP opening typically triggers a BAX-dependent apoptotic signaling, we next tested whether BAX inhibition can modulate 1-deoxySa toxicity[26]. The presence of a BAX inhibitor peptide (V5) protected HeLa cells from 1-deoxySa-induced cell death (Fig. 5f), indicating that BAX activation operates downstream of mPTP opening.

Taken together, these results suggest that mostly 1-deoxyDHCer (m18:0/24:1) but also 1-deoxyDHCer (m18:0/24:0) directly induce mitochondrial damage via mPTP opening and BAX activation, leading

**a**

**b**

**c**

**d**

to cell death. This defines a novel molecular mechanism of 1-deoxySL cytotoxicity, with potential implications for the understanding of the cellular basis of 1-deoxySL-associated diseases (Fig. 5g).

## Discussion

In this study, we identify a previously unrecognized molecular mechanism by which 1-deoxysphingolipids (1-deoxySLs) exert their cytotoxic effects. We demonstrated that mainly 1-deoxyDHCeramide (m18:0/24:1) and 1-deoxyDHCeramide (m18:0/24:0) structure is responsible for toxicity and mitochondrial dysfunction associated with an opening of the mPTP and BAX-mediated cell death. This finding establishes a direct link between 1-deoxySL structure and a defined cell death pathway, providing new mechanistic insight into the pathophysiology of 1-deoxySL-associated diseases.

**Fig. 2 | CRISPRi toxicity screen identifies gene candidates responsible for 1-deoxySphingolipids toxicity. a** Schematic representation of the CRISPRi-based toxicity screen. K562 CRISPRi-dCas9-KRAB cells were transduced with the CRISPRi v2 sgRNA library and cultured for five passages in the presence of 1-deoxySa (1.5 μM) or vehicle control. **b** Volcano plot displaying differentially enriched genes in 1-deoxySa-treated cells. The log fold change (logFC) indicates the difference in sgRNA as quantified by DNA sequencing. Genes sensitizing to 1-deoxySa toxicity (logFC≥0.5, q-value ≤ 0.01) are shown in red. **c** Schematic overview of the fatty acid elongation pathway showing the position of key genes identified in the screen.

*ELOVL1, HSD17B12, ACACA*, and *PTPLB* are involved in very-long-chain fatty acid (VLC-FA) biosynthesis, while *CERS2* incorporates VLC-FAs into ceramide and 1-deoxyDHCer species. **d** Gene set enrichment analysis (GSEA) of logFC values derived from the CRISPRi screen. Pathways were assigned using the KEGG database. Genes sensitizing cells to 1-deoxySa clustered within unsaturated fatty acid biosynthesis and elongation pathways, whereas protective genes were associated with diverse pathways including ribosome biogenesis, oxidative phosphorylation, DNA replication, RNA transport, and peroxisomal function. Illustrations were Created in BioRender. Hornemann, T. (2025) https://BioRender.com/ifocfkk.

1-deoxySLs are atypical sphingolipids generated via serine palmitoyltransferase (SPT) when alanine is utilized instead of serine. These lipids include long-chain bases (LCBs) such as 1-deoxysphinganine (1-deoxySa) and 1-deoxysphingosine (1-deoxySo), as well as their N-acylated derivatives (1-deoxyDHCer and 1-deoxyCer)[1,3]. The pathologically increased formation of 1-deoxySLs is implicated in a range of neurological conditions, including hereditary sensory and autonomic neuropathy type 1 (HSAN1), diabetic neuropathy, and macular telangiectasia (MacTel), as well as emerging roles in cancer metabolism and sarcopenia[5,9–13,17,27–30].

While previous studies have shown that 1-deoxySLs are toxic in various cell types, especially neurons, the molecular basis of this toxicity has remained unclear[19,20,31]. Previous studies have shown that inhibiting ceramide synthase by fumonisin B1 (FB1) protects against the cytotoxic effects of 1-deoxysphingolipids. For example, FB1 was reported to rescue primary neurons from 1-deoxySL-induced degeneration and to restore fibroblast migration inhibited by 1-deoxySL accumulation[20,22]. Consistent with these findings, we observed that FB1 treatment reduced the toxicity of 1-deoxySa. Mechanistically, FB1 inhibits the N-acylation of 1-deoxyLCBs, resulting in an accumulation of free 1-deoxySa and 1-deoxySo and a marked decrease in 1-deoxyDHCer and 1-deoxyCer levels. This supports the conclusion that N-acylated products, rather than the free LCBs, are responsible for toxicity.

In parallel, prior work has implicated fatty acid desaturase 3 (FADS3), an enzyme that introduces a Δ14(Z) double bond into sphingoid bases, as a modulator of 1-deoxySL toxicity. Specifically, FADS3 overexpression was shown to rescue cells from 1-deoxySa-induced toxicity, while FADS3 knockdown exacerbated toxicity, likely by favoring the accumulation of saturated, highly toxic 1-deoxyDHCer species[4]. In our study, we observed that 1-deoxySo, which is formed from 1-deoxySa by FADS3, is converted exclusively to 1-deoxyCer and exhibits substantially lower toxicity than 1-deoxySa. These data, together with the protection by FB1, showed that 1-deoxyDHCer, but not 1-deoxySa or 1-deoxyCer, represents the principal cytotoxic metabolites within the 1-deoxySL pathway.

CRISPR-based genetic screens have emerged as a powerful and widely adopted tool in functional genomics[32,33]. Unlike RNAi-based methods, CRISPRi achieves robust and specific gene silencing with reduced off-target effects, enabling the discovery of essential pathways and potential therapeutic targets in toxicity models. To systematically identify genetic regulators of 1-deoxySL toxicity, we employed a CRISPR interference (CRISPRi) screen in K562 cells. Our screen identified five high-confidence hits: *ELOVL1, HSD17B12, ACACA, PTPLB*, and *CERS2*—all of which are involved in fatty acid elongation or ceramide biosynthesis. These enzymes regulate the formation and utilization of very-long-chain fatty acids (VLC-FAs), particularly 24-carbon saturated and monounsaturated species, which are subsequently incorporated by CERS2 into sphingolipids[34]. Notably, VLC-1-deoxyDHCer species such as m18:0/24:0 and m18:0/24:1 predominate in cells while canonical sphingolipids display a broader distribution of the N-acyl chain length (data not shown).

Functional validation using siRNA and CRISPR-Cas9 knockout models of *ELOVL1* and *CERS2* confirmed their essential role in mediating 1-deoxySL toxicity. Disruption of either enzyme significantly

decreased VLC-1-deoxyDHCer formation and rescued cells from 1-deoxySa-induced toxicity, highlighting the critical importance of N-acyl chain length in determining cytotoxicity. These results are in agreement with prior yeast models showing a chain-length–dependent toxicity of 1-deoxySLs[35].

To explore the therapeutic potential of this pathway, we employed the preclinically validated ELOVL1 inhibitor compound (22) that was originally developed in the context of X-linked adrenoleukodystrophy (X-ALD)[24,25]. Compound (22) has been shown to be pharmacologically active in mouse and rat models, with established preclinical safety and efficacy in cynomolgus monkeys[24,25]. Treatment with (22) blocked VLC-FA elongation and thereby suppressed the formation of toxic VLC-1-deoxyDHCer species. This resulted in robust protection from 1-deoxySL-induced cytotoxicity in HeLa cells and neurotoxicity in primary dorsal root ganglion (DRG) neurons. Importantly, we validated the metabolic activity of (22) in vivo using a chicken embryo DRG, suggesting its action also in a neuronal context.

To further investigate the structure–toxicity relationship, we selectively stimulated the formation of individual 1-deoxyDHCer species in cells by supplementing defined LCFAs or VLCFAs in the presence of (22) to avoid further metabolism of the added FAs by ELOVL1. Without ELOVL1 inhibition, supplemented fatty acids such as palmitic acid (16:0) undergo elongation to C24:0 and C24:1 species, leading to mixed formation of multiple 1-deoxyDHCer species and therefore would complicate the interpretation of structure–toxicity relationships. Among all tested species, 1-deoxyDHCer (m18:0/24:1)—derived from nervonic acid (24:1)—was the most toxic, followed by m18:0/24:0 and m18:0/16:0. These findings point to nervonyl-1-deoxyDHCer (m18:0/24:1) as the main mediator of 1-deoxySL cytotoxicity. Of note, nervonic acid is enriched in myelin, suggesting a potential link between 1-deoxySL metabolism and demyelinating phenotypes observed in HSAN1 and related neuropathies.

Due to systemic mutations in SPT, 1-deoxySa are elevated in many tissues of HSAN1 patients. Nevertheless, the disease specifically affects sensory neuron function, leading to impaired pain and temperature sensation. Interestingly, previously reported single-cell RNAseq data of neurons[36] indicate a co-expression of *ELOVL1* and *CERS2* in specific nociceptor subtypes that also express pain receptors such as *TRPV1* or *TRPA1* (Supplementary Table 1). Based on our observations, these neuronal subtypes are likely more susceptible to 1-deoxySL toxicity, which might also explain the specificity of clinical symptoms seen in HSAN1, such as the loss of pain and temperature sensation. However, without further validation, it is still speculative, and interventional studies are needed to confirm this association in more detail.

Previous reports have suggested that 1-deoxySLs localize to mitochondria, where they disrupt mitochondrial morphology[19]. Currently, it is not clear what exact biochemical or biophysical mechanisms relate to the differential toxicities of the individual 1-deoxySL species. For canonical Cer, it has been shown that LC and VLC-Cer have opposing effects on $Ca^{2+}$-induced mitochondrial swelling[37]. Moreover, the N-acyl length and saturation of Cer have been shown to influence the biophysical properties of artificial membranes[38]. We confirmed these findings using Seahorse experiments, fluorescence microscopy,

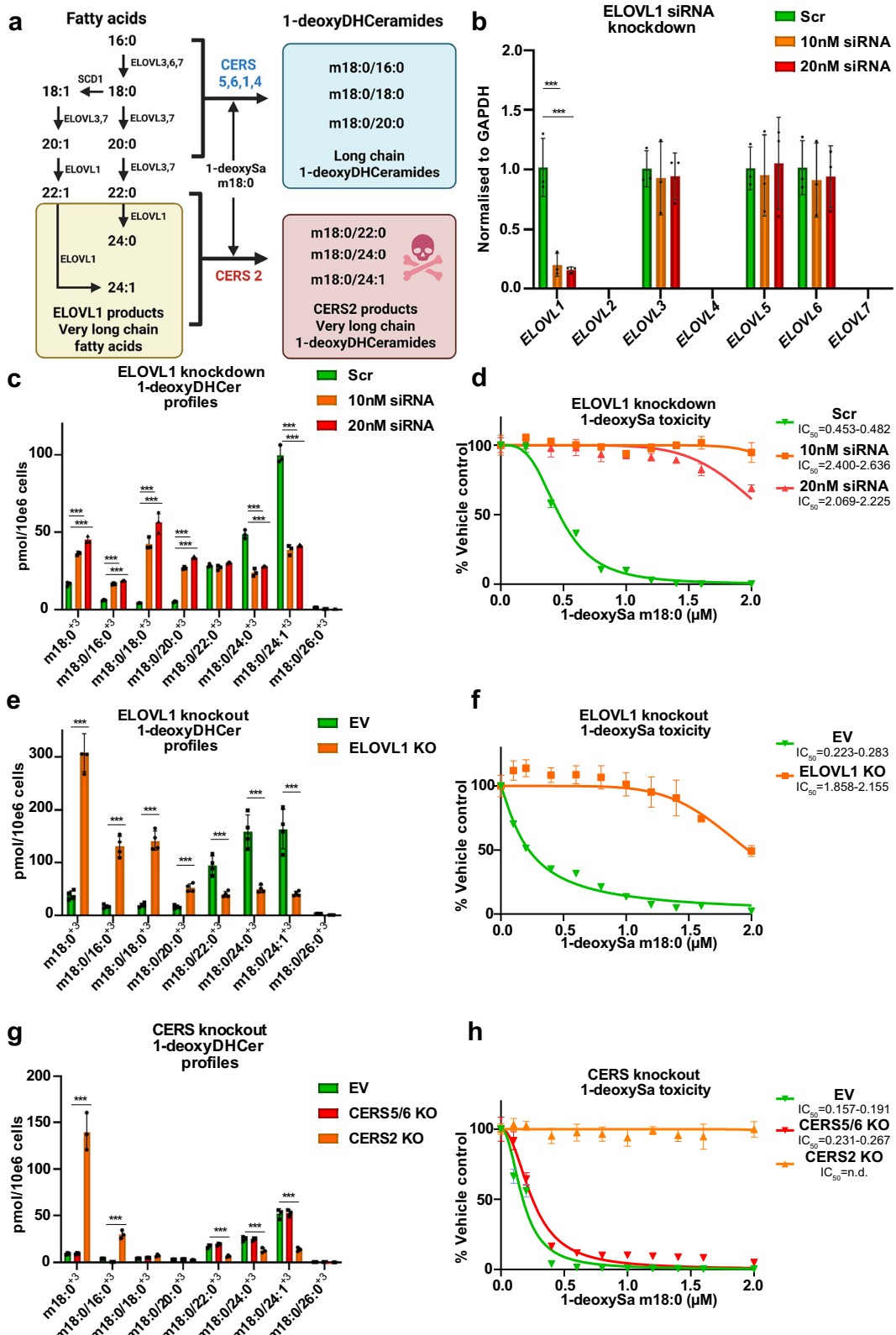

and mitochondrial swelling assays. Supplementation with 1-deoxySa induces loss of respiratory and glycolytic capacity and mitochondrial swelling. Notably, Nervonyl-1-deoxyDHCer (m18:0/24:1) caused the strongest mitochondrial swelling and cellular toxicity, which could be blocked by cyclosporin A (CsA), implicating mPTP opening as the underlying mechanism. Furthermore, inhibition of BAX, a downstream effector of mPTP-mediated apoptosis[26], rescued 1-deoxySa-induced

cell death. These findings define a new mechanism of 1-deoxySL toxicity involving mitochondrial destabilization via mPTP and BAX activation.

Despite these insights, our study has certain limitations. The efficacy of (22) has not yet been tested in an animal model of 1-deoxySL-associated diseases such as HSAN1 or diabetic neuropathy. Importantly, while (22) was designed to cross the blood–brain barrier

**Fig. 3 | Genetic targeting of *ELOVL1* and *CERS2* confirms their role in 1-deoxysphingolipids mediated toxicity. a** Schematic overview of the metabolic pathway linking fatty acid elongation by ELOVL1-7 to 1-deoxyDHCeramide synthesis by ceramide synthases (CERS1-6). **b** mRNA expression levels of *ELOVL1–7* in HeLa cells following siRNA-mediated knockdown of *ELOVL1*. Silencing *ELOVL1* did not affect expression of other *ELOVL* isoforms. Expression levels were determined using RT-qPCR and normalized to *GAPDH* ($n = 3$, independent biological replicates). **c** Quantification of isotopically labeled 1-deoxyDHCer (m18:0/xx:y)$^{+3}$ species in HeLa cells after siRNA *ELOVL1* knockdown and subsequent d$_3$-1-deoxySa supplementation. **d** Toxicity of 1-deoxySa in HeLa cells transfected with scrambled or *ELOVL1*-targeting siRNAs (10 or 20 nM), 72 h prior to treatment. **e** Quantification of isotopically labeled 1-deoxyDHCer species (m18:0/xx:y$^{+3}$) in CRISPR Cas9-generated *ELOVL1* KO and control HeLa cells (EV, Empty vector treated) after d$_3$-1-deoxySa supplementation **f** 1-deoxySa-induced toxicity in *ELOVL1* KO and control HeLa cells (EV). **g** Abundance of d$_3$-1-deoxyDHCer species (m18:0/xx:y$^{+3}$) in CRISPR

Cas9 *CERS2* KO, *CERS5/6* KO, and EV control HeLa cells following d$_3$-1-deoxySa supplementation. **h** Toxicity of 1-deoxySa in *CERS2KO*, *CERS5/6KO* and Control (EV) cells. Lipid species were quantified by untargeted high-resolution LC-MS/MS and normalized to internal standards and cell counts. Bar graphs are presented as mean ±SD ($n = 4$ biologically independent replicates). Significance was determined using two-tailed unpaired Student's t-test with multiple testing correction (two-stage step-up method of Benjamini, Krieger, and Yekutieli). Reported p.adjusted values: $p < 0.05$ (*), $p < 0.01$ (**), $p < 0.001$(***). Exact *p*-values and statistical source data are provided in Source Data 1. Toxicity data were measured by the CellTiter-GLO assay and normalized to vehicle-treated controls and shown as mean±SD ($n = 4$, ($n = 4$, biologically independent replicates, defined as separately cultured and treated HeLa cell populations from independent transfections or gene editing procedures). IC$_{50}$ values were calculated by nonlinear regression with 95% confidence intervals. Illustrations were created in BioRender. Hornemann, T. (2025) https://BioRender.com/ifocfkk.

for treating X-ALD[24,25], this property may be less desirable for targeting peripheral neuropathies. Future work should therefore focus on developing ELOVL1 inhibitors optimized for peripheral selectivity and tested in PN-relevant disease models.

In summary, our findings establish a direct link between the chemical structure of 1-deoxySLs and their cytotoxicity, revealing that both the saturation of the long-chain base and the length and unsaturation of the N-acyl chain determine toxicity. Most importantly, we identify 1-deoxyDHCer (m18:0/24:1) as a specific, mitotoxic species and uncover mPTP/BAX-associated mitochondrial dysfunction as a new molecular mechanism of 1-deoxySL-induced cell death. These findings support ELOVL1 inhibition as a promising therapeutic strategy for 1-deoxySL-associated diseases, including HSAN1, MacTel, and diabetic neuropathy.

## Methods

### Cell culture
HeLa cell lines were grown in high-glucose Dulbecco's Modified Eagle's Medium (DMEM, Thermo Fisher Scientific) supplemented with 10% fetal bovine serum (FBS) and 1% Penicillin/Streptomycin (P/S). K562 human myeloid leukemia cells were cultured in RPMI1640 Medium GlutaMAX (Thermo Fisher Scientific) supplemented with 10 % FBS and 1% P/S. All cell lines were kept in the incubator at a 5% CO$_2$ atmosphere and 37 °C. Cells were tested for Mycoplasma contamination.

HeLa CERS2 KO, CERS5/6 KO, and ELOVL1 KO were prepared as described previously[39].

### Cultures of dissociated DRG neurons
Dissociated dorsal root ganglia (DRG) neurons were cultured as described previously[40]. Briefly, DRGs of 9-day-old chicken embryos were dissected and collected in cold PBS. DRGs were centrifuged at 1000 rpm for 5 minutes at room temperature. PBS was exchanged with 0.25% Trypsin in PBS (15090-046, Invitrogen) containing DNase (final concentration: 0.2%; 10104159001, Roche) and incubated for 20 minutes at 37 °C. DRGs were collected by centrifugation, suspended in 1 ml DRG medium, and triturated with a fire-polished glass pipette with a diameter of about 0.5 mm. DRG media consisted of MEM with Gluta-max (41090-028, Invitrogen), Albumax (4 mg/ml, 11020-021, Invitrogen), N3, and NGF (20 ng/ml, 13290-010, Invitrogen). Per well, 20000 cells were plated in 8-well Lab-Tek plates (177445, Nunc) and incubated at 37 °C with 5% CO$_2$. After initial growth, 1-deoxySa and/or ELOVL1 inhibitor (compound 22 (22), HY-145272, MedChemExpress) were added to the corresponding wells. Cells were fixed after 40 hours of incubation with 4% PFA and washed 3 × 10 minutes with PBS.

### ELOVL1 inhibition and d3-1-deoxySa flux in chicken embryo-derived dorsal root ganglia
To investigate whether ELOVL1 inhibition by (22) could influence the 1-deoxySL profiles, chicken embryos were utilized, as previously

described[41,42]. Briefly, 4-days-old (E4) chicken embryos were windowed, and d3-1-deoxySa (50 μM, 50 μL) was injected to the embryo with or without (22) (2.5 μM, 50 μL). After 4 days, 7-days-old chicken DRGs were dissected according to previously established protocols[40]. For DRG dissection, embryos were pinned to a dissection plate, and the internal organs, ventral vertebrae, and spinal cord were carefully removed to expose the DRG. The DRGs were then collected in PBS, and then excess PBS was discarded. The tissues were subsequently frozen at −20° until lipid extraction and mass spectrometry analysis as described below.

### Silencing of ELOVL1
siRNAs targeting human ELOVL1 (s34994, Thermo Fisher Scientific) were used to silence (knockdown) ELOVL1 expression. ELOVL1 targeting siRNA or non-coding negative control Scramble (Scr, SR30004, OriGene) siRNAs were diluted to a final concentration of 10 nM or 20 nM in reduced-serum media (Opti-MEM, Gibco). Transfection was performed using Lipofectamine RNAiMAX transfection Reagent (Thermo Fisher Scientific) according to the manufacturer's recommendations. The media was replaced after 24 hours with fresh DMEM (10%FBS), and cells were grown in total for 72 hours before the start of labeling experiments. Knockdown efficiency and influence on other ELOVL isoforms were determined using qRT-PCR as described below.

### qRT-PCR
For qRT-PCR, cells were harvested using trypsinisation as described above. Cell pellets were lysed using TRIzol reagent, and RNA was extracted using the phenol/chloroform method following the manufacturer's instructions. RNA was purified using Ethanol, and concentration/purity was determined using NanoDrop (Thermo Fisher Scientific). Reverse transcription was performed using Maxima Reverse Transcriptase following the manufacturer's instructions (EP0742, Thermo Fisher Scientific).

qPCR reactions were performed using SYBR Green qPCR mastermix. Absolute concentration of mRNA was calculated from the dilution curve (1/10, 1/100, 1/1000, and 1/10,000 dilutions) of an adequate plasmid. GAPDH was used as an in-house gene loading control.

### Stable isotope labelling assay
For the SL labelling assay, cells were plated at 200,000 cells/mL in 6-well plates. Cells were grown for 48 hours to 70% confluency. 24 hours before harvesting, the medium was replaced with DMEM/10% FBS/1%P/S containing d$_3$-1-deoxysphinganine (860474, Avanti Polar Lipids) and additional treatments (compound 22 (22), HY-145272, MedChemExpress; FB1, F1147, Sigma-Aldrich) as indicated in the figures. Cells were harvested by trypsinisation and counted using Beckman Coulter Z2 (Beckman Coulter). Next, cells were centrifuged at 850 g at 4 °C, and cell pellets were washed 2 times with cold PBS. Cell pellets were then frozen and kept at −20 °C until further analysis.

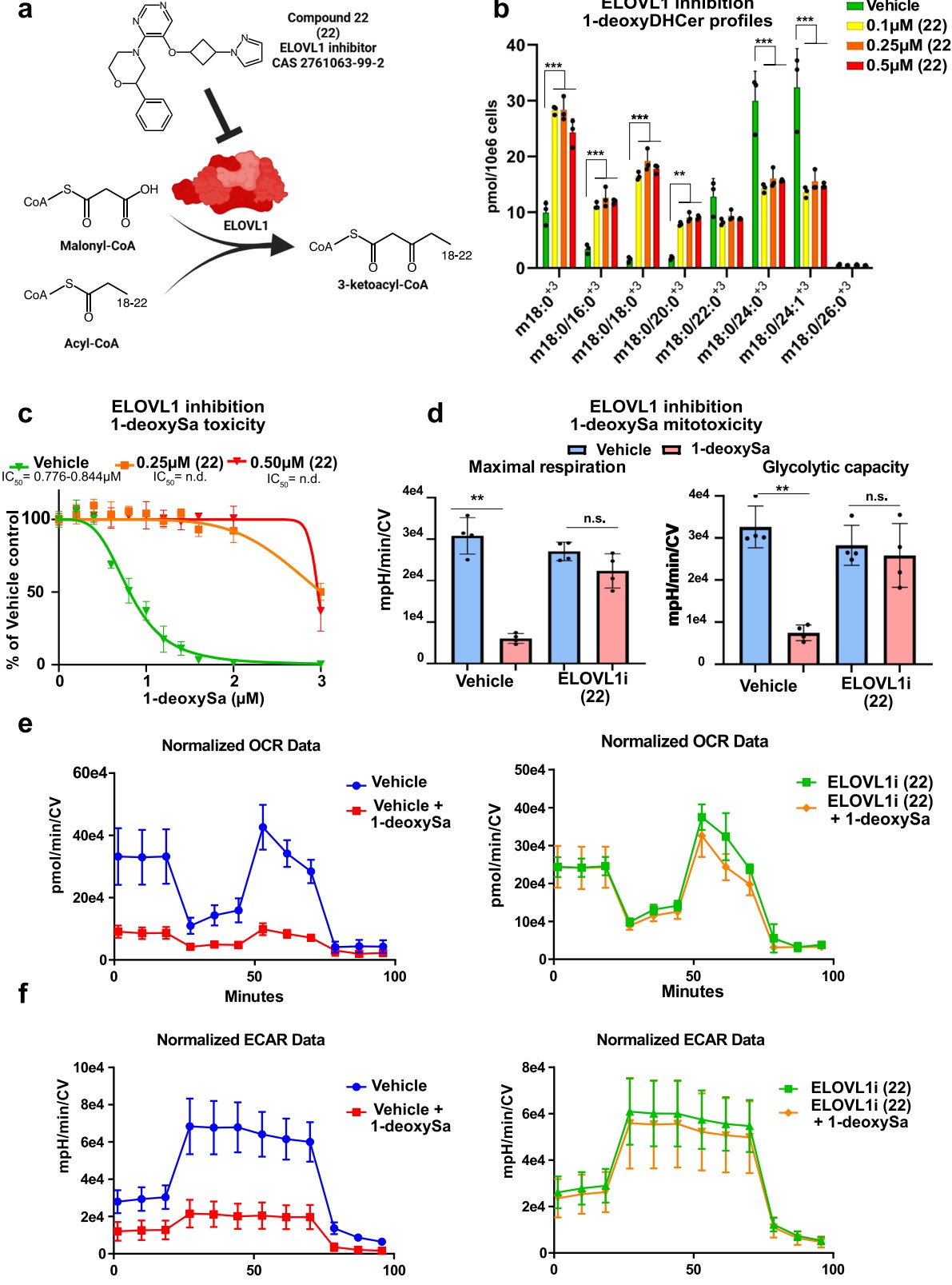

## Fatty acids supplementation

Fatty acids were supplemented to cells as sodium salts conjugated to Bovine Serine Albumin (BSA). First, 10 μmol of fatty acids were weighed into an Eppendorf tube, and EtOH (100 μL), $H_2O$ (300 μL), and 0.1 M NaOH (100 μL) were added. Next, the mixture was heated at 75 °C and mixed for 2 hours (Thermomixer (Eppendorf)). Afterwards, 500 μL of $H_2O$ was added (yielding a total 10 mM

concentration). BSA complexes were prepared by the addition of 100 μL of fatty acid sodium salts to 900 μL BSA (Essentially fatty acid-free (Thermo Fisher Scientific)) in DMEM (6.66 mg/mL) and shaking at 30 °C for 1 hour ((Thermomixer (Eppendorf)) and used directly for isotope labelling as described above. For vehicle control, no fatty acids were added, and the whole process was performed accordingly.

**Fig. 4 | Pharmacological inhibition of ELOVL1 mitigates 1-deoxysphingolipids induced mitochondrial and cellular toxicity. a** Schematic enzymatic reaction catalyzed by ELOVL1 and its inhibition by the small-molecule inhibitor Compound 22 (ELOVL1i; (22); CAS 2761063-99-2). **b** Quantification of 1-deoxyDHCer (m18:0/xx:y[+3]) species in HeLa cells supplemented with d$_3$-1-deoxySa and ELOVL1 inhibitor (22) or vehicle control ($n = 4$, biological replicates). **c** Cytotoxicity of 1-deoxySa in HeLa cells treated with ELOVL1i (0.25 μM) or vehicle. Data were normalized to vehicle-treated controls and shown as mean±SD ($n = 4$, separately cultured and treated HeLa cell populations). IC$_{50}$ values were derived from nonlinear regression analysis with 95% confidence intervals. **d** Maximal respiration and glycolytic capacity of HeLa cells treated with vehicle, 1-deoxySa (0.5 μM), or 1-deoxySa+ELOVL1i (0.25 μM), measured using a Seahorse XF analyzer. **e** Oxygen consumption rate (OCR) and **f** Extracellular acidification rate (ECAR) over time in vehicle- or ELOVL1i (22)-treated HeLa cells following exposure to 1-deoxySa (0.5 μM) measured using a Seahorse XF analyzer. All Seahorse data were normalized to crystal violet staining (CV) performed after the assay to correct for cell number and viability. This normalization was essential, as 1-deoxySa-induced cytotoxicity significantly affects viable cell count and could confound metabolic measurements. Data are shown as mean±SD ($n = 4$, individual biological replicates). Significance was determined using two-tailed unpaired Student's t-test with multiple testing correction (two-stage step-up method of Benjamini, Krieger, and Yekutieli). Reported p.adjusted values: $p < 0.05$ (*), $p < 0.01$ (**), $p < 0.001$(***). Exact $p$-values and statistical source data are provided in Source Data 1. Illustrations were created in BioRender. Hornemann, T. (2025) https://BioRender.com/ifocfkk.

## Lipid extraction and analysis

Lipidomics analysis was performed as described previously[43]. Extraction was performed by mixing (Thermomixer (Eppendorf), 60 minutes, 37 °C) the cell pellets, plasma (100 μL), or tissue homogenate with extraction buffer consisting of a mixture: methanol: methyl *tert*-butyl ether: chloroform 4:3:3 (v/v/v) and internal standards. After centrifugation (31,900 g, 10 minutes, 37 °C), the single-phase supernatant was collected, dried under N2, and stored at −20 °C. Before analysis, lipids were dissolved in 100 μL of MeOH Thermomixer (Eppendorf), 60 minutes, 37 °C) and separated on a C30 Accucore LC column (Thermo Fisher Scientific, 150 mm × 2.1 mm × 2.6 μm) or C18 ACQUITY UPLC CSH (Waters, 150 mm × 2.1 mm × 1.7 μm) using gradient elution with A) Acetonitrile: Water (6:4) with 10 mM ammonium acetate and 0.1% formic acid and B) Isopropanol: Acetonitrile (9:1) with 10 mM ammonium acetate and 0.1% formic acid at a flow rate of 260 μL/minute using Transcend UHPLC pump (Thermo Fisher Scientific) at 50 °C. The following gradient was used: 0 min 30% (B), 0.5 min 30% (B), 2 min 43% (B), 3.3 min 55% (B), 12 min 75% (B), 18 min 100% (B), 25 min 100% (B), 25.5 min 30% (B), 29.5 min 30% (B). Eluted lipids were analysed by a Q-Exactive plus HRMS (Thermo Fisher Scientific) in positive and negative modes using heated electrospray ionisation (HESI, Sheath gas flow rate=40, Aux gas flow rate=10, Sweep gas flow rate=0, Spray voltage=3.50, Capillary temperature=320 °C, S-lens RF level = 50.0, Aux gas heater temperature=325 °C). MS2 fragmentation spectra were recorded in data-dependent acquisition mode with a top 10 approach and constant collision energy 25 eV. 140 000 resolution was used for full MS1 and 17 500 for MS2. Peak integration was performed with TraceFinder 4.1, Skyline daily. Lipids were identified by predicted mass (resolution 5ppm), retention time (RT), and specific fragmentation patterns using an in-house Lipidcreator and MSDIAL compound databases. Isotopic enrichment was tracked by M + 3 for d3-1-deoxySa-treated cells. Next, lipid concentrations were normalized to the corresponding internal standards (one per class) and cell number.

*List of internal standards*:

d5-1-deoxymethylsphinganine (m17:0, 860476, Avanti Polar Lipids) 100pmol/sample

1-deoxyDHCeramide (m18:0/12:0, 860460 P, Avanti Polar Lipids) 100pmol/sample

1-deoxyCeramide (m18:1/12:0, 860455, Avanti Polar Lipids) 100pmol/sample

DHCeramide (d18:0/12:0, 860635 Avanti Polar Lipids) 100pmol/sample

Ceramide (d18:1/12:0, 860512, Avanti Polar Lipids) 100pmol/sample

SM (d18:1/12:0, 860583, Avanti Polar Lipids) 100pmol/sample

Glucosylceramide (d18:1/8:0, 860540, Avanti Polar Lipids) 100pmol/sample

SPLASH 2.5 μL/sample (330707, Avanti Polar Lipids)

Transitions used for the identification of Sphingolipids and 1-deoxySphingolipids:

1-deoxyCeramides and 1-deoxyDHCeramides:

$[M+H]^+ \rightarrow [M+H-H_2O]^+$, $[M+H]^+ \rightarrow [M+H-H_2O-FA]^+$

Ceramides and dihydroCeramides:

$[M+H]^+ \rightarrow [M+H-H_2O]^+$, $[M+H]^+ \rightarrow [M+H-H_2O-FA]^+$,

$[M+H]^+ \rightarrow [M+H-2xH_2O-FA]^+$

Free atypical long-chain bases

$[M+H]^+ \rightarrow [M+H-H_2O]^+$

Note: FA represents corresponding fatty acyl.

## Toxicity assays

For the toxicity assay, cells were grown for 72 hours in the 96-well plates in DMEM (supplemented with 10% FBS and 1% P/S) with added treatments as indicated in the figures. All conditions were corrected for the solvent concentration. Chemicals used were dissolved in DMSO to a 1 mM concentration and added directly to the media.

Chemicals used: Fuminosin B1 (F1147, Sigma-Aldrich); Compound 22 (HY-145122, MedChemExpress); Cyclosporin A (HY-B0579, MedChemExpress); Bax inhibitor peptide V5 (HY-P0081, MedChemExpress), 1-deoxySa (860493, Avanti Polar Lipids).

The number of viable cells was determined by quantitation of the ATP present using CellTiter-Glo ® Luminescent Cell Viability Assay (G7570, Promega) according to the manufacturer's recommendations. The chemiluminescence signal was collected using a TECAN Infinite M 200 Pro reader. Data were normalized to the average of Vehicle-treated wells.

## Immunocytochemistry of DRG dissociated sensory neurons

Cells were permeabilized with 0.1% Triton-X 100 in PBS and blocked for 15 minutes with 5% FCS in PBS (blocking buffer) at room temperature. Primary antibodies were diluted in blocking buffer and added to the cells for 1 hour. Primary antibodies included anti-Neurofilament-M (1:2000; clone RMO270, RRID: AB_2315286, Invitrogen) and anti-TOMM20 (rabbit anti-TOMM20, HPA011562, 1:800, Sigma-Aldrich). Cells were washed with PBS and incubated with secondary antibodies diluted in blocking buffer for another hour. Then cells were stained with Hoechst (2.5 μg/ml in PBS, H3570) for 5 minutes and rinsed three times with PBS. Finally, cells were mounted with Mowiol/Dabco.

## Microscopy and quantification of neurite density

Neurons were imaged with a 20x air objective and an Olympus BX63 upright microscope. Neurite growth was analyzed with Fiji/ImageJ. For this purpose, a region of interest (6500 × 6500 micrometers) was chosen in the center of each well. Images were transformed into 8-bit and a threshold was applied. The total area of neurons and neurites was assessed with the "Measure" feature.

## Live cell microscopy

For the live cell microscopy, cells were grown in 96-well plates (Ibidi) and the indicated treatment was added by exchange of the media. Then, the plate was immediately transferred to a live-cell imaging microscope (Olympus IX81) with a motorized stage, fitted with an incubator with pre-heated humidified atmosphere (Ibidi mixer). Phase-contrast images were acquired at 20x magnification every hour for 48 hours. In all procedures, cells were kept at 5% CO$_2$ and 37 °C.

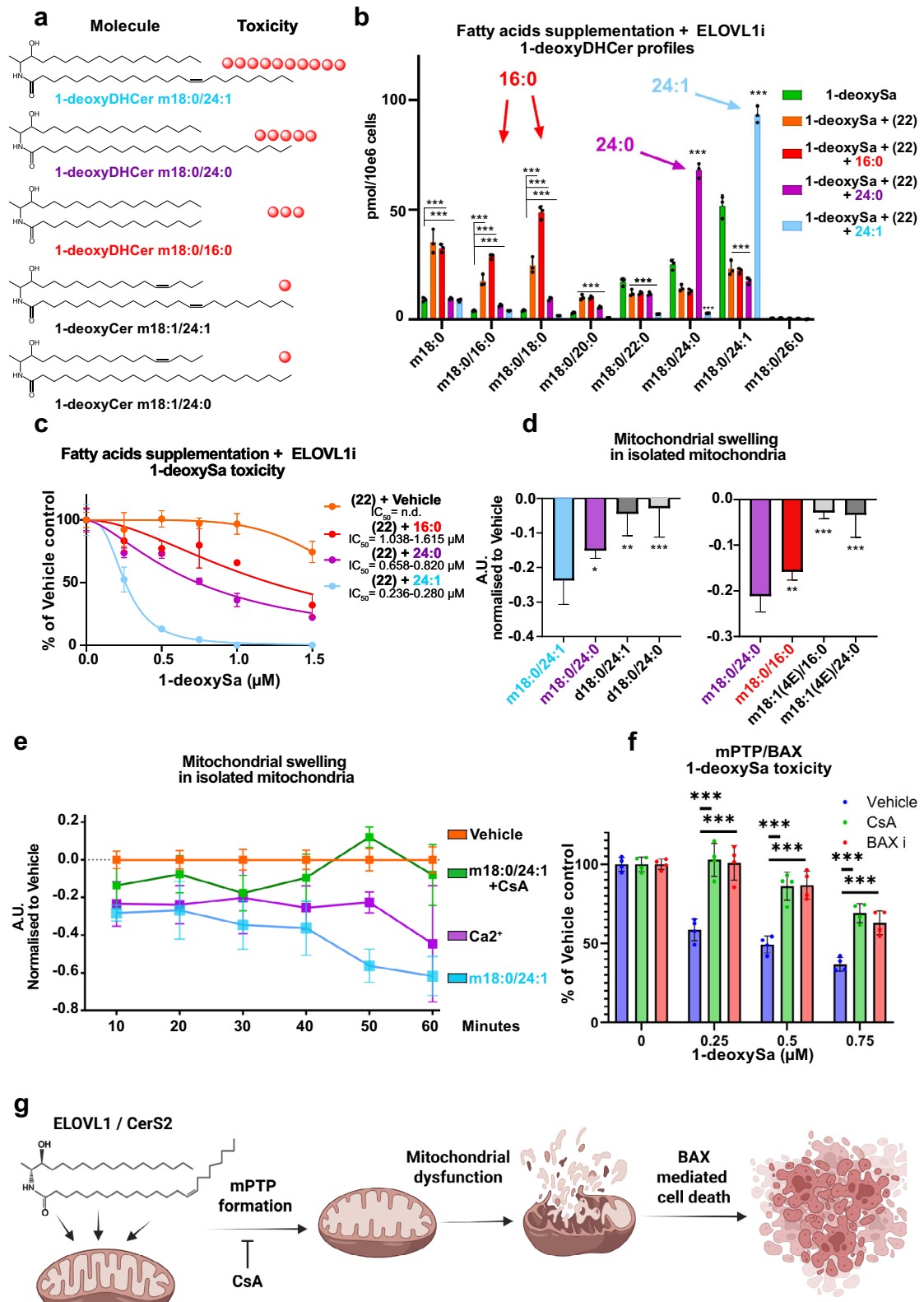

## Whole-genome CRISPRi 1-deoxySa toxicity screen

CRISPRi toxicity screen was performed using the K562 cell line as described in previously published screens[44,45]. Briefly, K562 cells stably expressing dCas9-KRAB were transduced with the CRISPRi v2 sgRNA library. This library comprises ~200,000 sgRNAs targeting ~20,000 protein-coding genes (10 sgRNAs per gene) and ~150,000 non-targeting sgRNAs grouped into ~15,000 pseudogenes, resulting in a total of ~350,000 sgRNAs (60% targeting, 40% non-targeting. 48 hours later, cells were selected with 0.75 µg/ml puromycin for 72 hours. Next, cells were recovered from the selection with puromycin-free media for 48 hours. Two independent biological replicates (200 million cells per replicate) were prepared for both untreated and 1.5 µM 1-deoxySa-treated conditions (860493, Avanti Polar Lipids). For each flask in the untreated and treated groups, the cells were kept at 0.5 million cells/ml

**Fig. 5 | Nervonyl-1-deoxyDHCeramide (m18:0/24:1) induces the strongest mitochondrial damage via mPTP and BAX, revealing a novel mechanism of 1-deoxysphingolipid cytotoxicity. a** Schematic representation of the chemical structures of individual 1-deoxysphingolipid (1-deoxySL) species and their relative cytotoxic potential. **b** Quantification of 1-deoxyDHCer species in HeLa cells supplemented with 1-deoxySa and co-treated with long-chain (LCFA, 25 μM) or very-long-chain fatty acids (VLCFA, 25 μM), in presence of ELOVL1 inhibitor Compound (22) (0.25 μM). Fatty acids were applied as sodium salts complexed with 4 mg/mL bovine serum albumin (BSA); fatty acids free BSA alone served as vehicle control (green bar, 1-deoxySa condition). Data are shown as mean±SD (*n* = 3, separately cultured and treated HeLa cell populations). **c** 1-deoxySa cytotoxicity in HeLa cells co-treated with LCFA or VLCFA (25 μM) in the presence of ELOVL1i (0.25 μM). Toxicity was normalized to the corresponding fatty acids free BSA vehicle-treated controls. Data represent mean±SD (*n* = 4, biological replicates). IC$_{50}$ values were derived from non-linear regression with 95% confidence intervals. **d** Mitochondrial swelling in freshly isolated mouse liver mitochondria after direct addition of individual 1-deoxyDHCer, 1-deoxyCer, or canonical dihydroceramide (DHCer) species, as indicated (*n* = 7, independently treated populations of isolated mitochondria). **e** Mitochondrial swelling in isolated liver mitochondria upon exposure to 1-deoxyDHCer (m18:0/24:1), in the presence or absence of the mitochondrial permeability transition pore (mPTP) inhibitor cyclosporin A (CsA, 10 μM). Positive control of 50 μM CaCl is displayed in purple (*n* = 6). **f** 1-deoxySa cytotoxicity in HeLa cells treated with CsA (10 μM) or Bax inhibitor (Bax i, Bax inhibitor peptide V5, 10 μM). Significance was assessed using two-tailed unpaired Student's *t*-test against 1-deoxySa + (22) condition, with correction for multiple testing (two-stage step-up method of Benjamini, Krieger, and Yekutieli). $p < 0.05$ (*), $p < 0.01$ (**), $p < 0.001$ (***). **g** Schematic illustration of our proposed model of 1-deoxySL toxicity. We hypothesize that accumulation of very-long-chain (VLC) 1-deoxyDHCer species- particularly 1-deoxyDHCer (m18:0/24:1)- triggers opening of the mitochondrial permeability transition pore (mPTP), resulting in mitochondrial swelling, dysfunction, and BAX-mediated cell death. Illustrations were created in BioRender. Hornemann, T. (2025) https://BioRender.com/ifocfkk.

---

daily. This means 1.5 μM 1-deoxySa caused a 50% growth inhibition. Treatment was continued until the untreated cells doubled five to eight times more than the treated cells. Genomic DNA was extracted using standard phenol-chloroform protocols. sgRNA sequences were PCR-amplified using staggered forward primers (NGS-Lib-Fwd-1 to -10) and a common reverse primer (NGS-Lib-Rev). PCR products were sequenced using Illumina HiSeq with 100 bp single-end reads, targeting an average read depth of >200 reads per sgRNA per replicate. sgRNA counts were aligned, normalized, and analyzed using the ScreenProcessing pipeline (https://github.com/mhorlbeck/ScreenProcessing), which applies median normalization, control-based noise modeling, and Mann-Whitney U tests to compute gene-level significance score.

### Mitochondria isolation from liver

Functional mitochondria were isolated from the livers of 10–12 week-old female C57/BJ mice (modification of the method described in ref. 46. Briefly, after excision, the livers were placed in an ice-cold isolation buffer (200 mM mannitol, 50 mM sucrose, 5 mM KH$_2$PO$_4$, 5 mM MOPS, 0.1% fatty acid free BSA, 1 mM EGTA, adjusted to pH 7.15 with KOH), homogenized with a motor-driven tightly fitting glass-Teflon Potter grinder operated at 1 600 rpm and centrifuged at 3 minutes (4 °C, 1 100 g). The supernatant was centrifuged at 4 °C for 10 minutes at 10,000 *g*. The pellet containing the mitochondrial fraction was resuspended in 200–400 μl of assay buffer (120 mM KCl, 10 mM Tris, 5 mM KH$_2$PO$_4$, pH 7.4) and centrifuged for 10 minutes (4 °C, 10,000 g). Supernatant was decanted, and the pellet was resuspended in the assay buffer.

Mitochondria were kept on ice, and an aliquot was used for the assessment of protein content by BCA assay (Thermo Fisher Scientific) according to the manufacturer's recommendations.

### Mitochondrial swelling assay

The activation of the mitochondrial permeability transition pore causes mitochondrial swelling, which is measured spectrophotometrically as a decrease in absorbance at 540 nm as described previously[37]. Briefly, the activation of the mitochondrial permeability transition pore causes mitochondrial swelling, which is measured spectrophotometrically as a decrease in absorbance at 540 nm. 250 μg aliquots of mitochondria suspension were distributed in a clear bottom 96-well plate, and then incubated directly with the indicated lipids: 1-deoxyDHCer (1-deoxyDHCer m18:0/16:0, Avanti Polar Lipids, 860462); (1-deoxyDHCer m18:0/24:0, Cayman Chemicals, 27017); (1-deoxyDHCer m18:0/24:1, Cayman Chemicals, 27562), 1-deoxyCer (1-deoxyCer m18:1(E)/16:0, Cayman Chemicals, 30029); (1-deoxyCer m18:1(E)/24:0, Cayman Chemicals, 30029) or DHCer (DHCer d18:0/24:0, Avanti Polar Lipids, 860628), (DHCer d18:0/24:1, Avanti Polar Lipids, 860629) (Ethanolic solutions) at $4 \times 10^{-3}$ nmol per ug of protein[37]. After 1 hour, the decline in the absorbance at 540 nm was measured. For the inhibition of mPTP, cyclosporine A (CsA, 10 μM) was used. CaCl2 (Ca2 + , 50 μM) was used as the positive control for the experiments. For the time series of measurement, the absorbance at 540 nm was recorded every 10 min after the indicated treatments until 1 hour.

### Seahorse assay

The Seahorse extracellular flux (XF) 24 analyzer (Agilent) was employed to analyze cellular oxygen consumption rate (OCR) and extracellular acidification rate (ECAR). Initially, 20,000 cells were seeded onto a Seahorse microplate containing 250 μL of growth medium containing treatment as indicated in the figures. 72 hours later, Seahorse XF basal medium was added (maintained at pH 7.4). The medium was maintained in a CO$_2$-free incubator at 37 °C for an hour. The subsequent mitochondrial stress test or glycolysis stress test was executed to quantify OCR and ECAR, respectively.

For the mitochondrial stress test, the XF basal medium was enriched with 10 mM glucose, 1 mM pyruvate, and 2 mM glutamine. OCR and ECAR measurements were taken during distinct phases: baseline, after adding oligomycin (final concentration 1 μM), following the introduction of carbonyl cyanide-p-trifluoromethoxyphenyl-hydrazone (FCCP; final concentration 1 μM), and subsequent to the mixture of rotenone/antimycin A addition (final concentration of 0.5 μM each).

Conversely, the glycolysis stress test was conducted using XF basal medium supplemented with 2 mM glutamine. OCR and ECAR observations were collected under the subsequent conditions: baseline, after adding glucose (final concentration 10 mM), following the introduction of oligomycin (final concentration 1 μM), and after adding 2-deoxyglucose (2-DG; final concentration 50 mM).

The data were analysed using Agilent's Wave software. Lastly, flux rates were normalized to the Crystal violet signal (measured by Tecan reader, 565 nm) after fixation with 4% paraformaldehyde (PFA, Thermo Fisher Scientific).

### Gene expression analysis in dorsal root ganglion (DRG) cells

Gene expression data in individual DRG cells, obtained from single-cell RNA-seq data originally published by Usoskin et al. was downloaded from http://linnarssonlab.org/drg/. The expression of each gene was illustrated as reads-per-million (RPM), and the maximal expression level for each gene of interest was calculated by averaging the three highest values across all analyzed cells. A cell expressing the gene at levels higher than 5% of the maximal level was considered positive for the gene, which is a similar criterion as Usoskin et al. with the exception that we also included non-neuronal cell types in our analysis. Within the 11 neuronal cell types and non-neuronal cells, we calculated the fraction of cells expressing each gene of interest. In addition, we

calculated the fraction of cells positive for both *ELOVL1* and *CERS2* in each cell type.

## Data analysis, statistics, writing, and figures

All the data analysis and figure preparations were performed using GraphPad Prism 9.5.1 and Excel. Statistical analysis was performed using GraphPad Prism 9.5.1. Gene set enrichment analysis was performed using WEB-based Gene SeT AnaLysis Toolkit (WebGestalt.org). Illustrations were made using Biorender.com webpage under a Lab-academic subscription with a publication license. Chemical structures were drawn using ChemDraw Ultra 12.0 and Biorender.com under a Lab-academic subscription with a publication license. ChatGPT 4o (OpenAI) was used to improve grammar and language clarity; no text or data were generated by the model. Final figures were prepared using Affinity Designer 2 under an institutional/academic license.

## Reporting summary

Further information on research design is available in the Nature Portfolio Reporting Summary linked to this article.

## Data availability

Source data are provided with this paper. All datasets supporting the findings of this study, including those underlying Figs. 1, 3, 4, 5, and supplementary Fig. 1, as well as additional lipidomics data, are provided in Source Data 1. Guide RNA sequences and processed results from the CRISPRi toxicity screen used in Fig. 2 are available in Source Data 2 and 3. The raw sequencing data were generated at a previous institution and are no longer available. Additional raw data and analysis files are available from the corresponding author upon request. Source data are provided with this paper.

## Code availability

The code for sgRNA count alignment, normalization, and analysis is available via GitHub (https://github.com/mhorlbeck/ScreenProcessing).

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

## Acknowledgements

This project was supported by the Swiss National Science Foundation (SNF 310030_215134) and by the SNF under the frame of the European Joint Program on Rare Diseases (EJP RD + SNF 32ER30_187505) and COST Action 19105-Pan-European Network in Lipidomics and EpiLipidomics (EpiLipidNET). THA. was supported by the NCCR Chemical Biology (185898 attributed to Howard Riezman). Additionally, THA was supported by the French Government (National Research Agency, ANR) through the "Investments for the Future" programs LABEX SIGNALIFE ANR-11-LABX-0028 and IDEX UCAJedi ANR-15-IDEX-01, the ATIP-Avenir program (CNRS/Inserm). We also would like to acknowledge Howard Riezman for his support on this project and Aaliyah Chaussin for her contribution in making the KO cell lines.

## Author contributions

A.M. performed all the experiments and analysis, interpreted the data and wrote the manuscript, G.K. supervised the study and interpreted the data, E.Y., M.S., R.D. performed the DRG sensory neurons preparations and analysis, J.L. and T.P. performed CRISPRi screen, E.M. performed the mPTP and BAX toxicity assays, T.HA., S.R.G., K.S. generated *CERS2KO*, *CERS5/6*dKO and *ELOVL1KO* cell lines, P.H. prepared fatty acid complexes, G.Z. performed mitochondrial swelling experiments, A.M. and E.Y. performed the experiments and revisions of the manuscript, T.H. supervised the study and revised the manuscript.

## Competing interests

The authors declare no competing interests.
