## [Peer Review file · Nature Communications]

Very long chain fatty acids drive 1-deoxySphingolipid toxicity

Corresponding Author: Professor Thorsten Hornemann

Version 0:

Reviewer comments:

Reviewer #1

(Remarks to the Author)

This study provides a significant advance on our understanding of how toxic deoxysphingolipid species linked to diabetic polyneuropathy, MacTel, and HSAN1 mediate toxicity in disease. The authors employ an impressive combination of screens and assays to tease apart the specifics of a very complicated pathway. The CRISPRi screen and follow up analysis of the sphingolipid sub-species were very convincing. I have a couple comments regarding the mitochondrial toxicity assays and the hypothesis that ELOVL1/CERS2 co-expression explains cell specificity that I detail below. Overall this manuscript was well done and should be considered with revisions.

Specific comments:

1. For the silencing of CERSs and ELOVL1 using RNAi I did not see any data showing the specificity of the knock downs. Since both gene families have multiple genes with homologous sequences it would be appropriate to know the amount, if any, of crossover effect of each RNAi set on genes within the respective families. Some crossover is fine but we need to know the amount to make strong conclusions.
2. It is unclear from the Seahorse experiments whether the effects on OCR and ECAR are due to direct effects on mitochondrial function or because the cells are dying and are undergoing apoptosis. The seahorse experiments are performed with 0.5uM lipid at 3 days which is when the authors can see up to 50% loss in cells in their survival assays. This seems to be consistent with how dramatic the effect of 1-deoxySa on background levels of OCR/ECAR following the rotenone/antimycin A injection in extended figure 1A, B. A complete loss of background ECAR would suggest that the cells are just dying and that's why you're seeing a loss of function. Can the authors find a change in mitochondrial function prior to death, either with a lower dose of 1-deoxySa or at an earlier time point before cell loss?
3. The hypothesis that ELOVL1/CERS2 co-expression explains cell specificity is very interesting. If this were true, could you test it by looking for cell specific death in your DRG cultures following 1-deoxySa treatment?

Reviewer #2

(Remarks to the Author)

This study investigates the mechanisms underlying the toxicity of 1-deoxysphingolipids (1-deoxySL), atypical sphingolipids associated with sensory neuropathies. Through a CRISPRi toxicity screen in K562 cells, the researchers identified ELOVL1 and CerS2 as key factors mediating 1-deoxySL toxicity. Pharmacological inhibition of ELOVL1 prevented 1-deoxySL-mediated cellular and neuronal toxicity in DRG cells and SY5Y cells. The study also demonstrates that 1-deoxySL toxicity is linked to mitochondrial dysfunction and depends on the length and saturation state of both the long-chain base and the conjugated N-acyl chain, with Nervonyl-1-deoxyDHCeramide identified as the main mediator of toxicity.

Major points:

The toxicity phenotype of 1-deoxySL was established in HeLa cells (Fig. 1), but the screen was performed in K562 cells. To justify the choice of cell line, dose, and treatment duration for the screen, a similar toxicity phenotype should be demonstrated in K562 cells.

The authors suggest that the co-expression of ELOVL1 and CerS2 in specific DRG cell subtypes may explain the selective toxicity of 1-deoxySA in HSAN1 patients. To support this hypothesis, in vivo toxicity experiments combined with single-cell analysis would be beneficial.

The description of the screen methodology and analysis is incomplete and unclear. Essential information is missing, including number of cells used, library coverage, treatment dose and duration, number of replicates, raw sequencing data and processed screening results.

Crucially, the screen was CRISPRi-based, which inhibits gene expression at the transcriptional level rather than introducing mutations. However, the analysis described in the methods section is based on determining mutation frequencies, which is inappropriate for CRISPRi. Please clarify the analysis method used and ensure it aligns with the CRISPRi approach.

Minor points:

Scale bars are missing for Figure 4E.

Font sizes in figures are inconsistent and sometimes too small to read. Please standardize font sizes throughout all figures for improved readability.

Reviewer #3

(Remarks to the Author)

In the manuscript by Majcher et al. the authors demonstrate a so far unacknowledged role of ELOVL1 and CerS2 for 1-deoxySL toxicity. Both enzymes are involved upstream of the 1-deoxyDHCer synthesis. They contribute one building block of 1-deoxyDHCer, the VLC-FA, yielding ultimately VLC-1-deoxyDHCer, if deoxy-LCB are also in supply. While it has been discussed before, that the N-acylated 1-deoxySL might mediate the toxicity of 1-deoxySL, no published data directly shows an involvement of ELOVL1 and CerS2. As both enzymes have a long-established function in FA elongation and ceramide synthesis, their role for VLC-1-deoxySL synthesis might have been expected. The finding that the VLC-members are primarily responsible for the toxicity of 1-deoxySL is intriguing and at the heart of this manuscript. It opens up the possibility to influence the detrimental process in affected patients.

The methodology applied in the study is sound. The work meets the expected standards in the field.

In its current form the manuscript shows formal shortcomings. It appears hastily assembled from a previous submission elsewhere, without final proofing of the new version. An Extended Figure 1 is available, but not called in the text; Extended Figures 2 and 3 are not existing; Extended Figure 4 is called but not available. Confusing. A revision, including further experiments is needed.

Importantly, data on a more complex disease model such as HSN1 or diabetic mice would considerably strengthen the manuscript and appear necessary for publication in this journal.

The 'Abstracts' suffers from overstatements and should be toned down, e.g. in line26: While the manuscript establishes a role for CerS2 and ELOVL1, their role as 'key enzymes' is currently not fully substantiated. Please omit "the key enzymes" from this sentence.

Also, can the authors substantiate the claim in line31 that "this toxicity depends on the length ... of the long-chain base"?

Other points include:

L61: has been

L72: bad call of figure - Fig. 1A does not match the meaning of this sentence

L79: bad call of figure - Fig. 1A does not match the meaning of this sentence

Fig. 1B: How do these curves look for HeLa cells overexpressing FADS3?

L86: introduce line break

L88: No information provided on the concentration of 1-deoxySA during screening. How does the 1-deoxySA susceptibility of K562 cells compare to HeLa?

L95: check grammar

L98: Where would CerS5 and CerS6 be located?

L100/1/2: biosynthesis or the elongation; ribosomes; oxidative

L134: omit "a"

L136: no line break

L141: m18:0/18:0 does not appear to be changed

L174: Extended Figure 4 is not available – you may not call Extended Figure 4 before calling Extended Figures 1-3 in the text

L170+175: Are refs 19 and 27 the same?

L179: Extended Figure 4 is not available – you may not call Extended Figure 4 before calling Extended Figures 1-3 in the text

Fig. 4D: What does one data point represent? One cell, one analyzed image, ...

Fig. 4E: Panels e-l need to be presented as greyscale images. The TOMM20 signal appears to be diffuse and mitochondria are hard to discern – is this an image compression artifact? Please make sure the arrows point directly(!) to mitochondria.

Show larger enlargements.

L191: "significantly" - What quantification and statistical analysis are you referring to?

L193/5: Extended Figure 4 is not available – you may not call Extended Figure 4 before calling Extended Figures 1-3 in the text; Does this manuscript contain data on SH-SY5Y cells? If not, please revise also M&M.

L204: and glycolytic capacity

L213: The mentioned scale bars are missing in Fig. 4E.

L238: Where in M&M can this protocol be found? How is the solubility of ceramides secured?

Fig. 5A: doxicity > toxicity

Fig. 5B,C: How would these panels look in the absence of inhibitor 22? How in the presence of the pan-CerS-inhibitor FB1 instead of inhibitor 22?

Fig. 5D and L283: What exactly is the "indicated treatment"? No specifics given in M&M either. Please explain the assay and give reference (if applying): Why is there a decline in absorbance 540nm? What were the controls?

L267: add "in"

L288: Move caption to the top of the table. Add reference to previously published data (if applying).

Extended Figure 1 is not called in the text and appears a fragment. Is this Extended Figure 4? If so, the fragmentation of mitochondria in panel C cannot be recognized (potentially due to image compression). Please provide suitable images and some sort of quantitation of the phenomena. The desired information in panel D is lost altogether (potentially due to image compression). Please provide suitable images of the phenomena.

Discussion: Revise and sharpen. Avoid overly borrowing text passages from earlier parts of the manuscript. Please check journal style on calling figures in the 'Discussion'.

L349/50: Where is this data to be found (see my comment on Fig. 5B,C above)?

L363: "Retraction from axons"- Do you show data on this? Is this different from the data on 'neurite area' in Fig. 4D?

Please check journal style on capitalization: L-Alanine or L-alanine; Serine-palmitoyltransferase or serine-palmitoyltransferase etc. in 'Abstract' and elsewhere.

Please clarify author contribution: It appears puzzling that MA performed all(!) the experiments and analysis, interpreted the data and wrote the manuscript but is not an author, while the manuscript has two joint-first-authors, whose contribution is unclear.

Version 1:

Reviewer comments:

Reviewer #1

(Remarks to the Author)

The authors have addressed all of my concerns.

minor not there is a typo on page 17 "This demonstrates that 1-deoxyDHCer (m18:0/24:1) directly perturbs mitochondrial integrity." should read "integrity"

Reviewer #2

(Remarks to the Author)

I still have the following points regarding the CRISPRi screen in the revised manuscript:

1. In the Results section, the authors state: "Haploid K562 cells expressing CRISPRi-dCas9-KRAB were transfected." However, K562 cells are generally considered diploid, albeit with chromosomal abnormalities. Please correct this statement or provide clarification regarding the haploid status of the K562 cells used.
2. In the legend for Figure 2b, the authors state: "the difference in sgRNA as quantified by RNA sequencing." However, sgRNA abundances in CRISPR screens are typically quantified by DNA sequencing, not RNA sequencing. Please revise this description to accurately reflect the methodology used.
3. There is no legend provided for Figure 2d. Please add an appropriate figure legend for this panel.
4. In the Methods section, the authors mention: "10 computer-generated, non-targeting sgRNAs clustered into 15K 'pseudogenes'." It is unclear how 10 non-targeting sgRNAs could generate 15,000 "pseudogenes." Please provide a more detailed explanation regarding this approach.
5. The sentence: "In all, 350K sgRNAs were used with a 60% targeting, 40% non-targeting ratio." requires further clarification. Please elaborate on the composition of the sgRNA library and the rationale for this ratio.
6. The response letter states that information on "the number of biological replicates" and "details regarding sequencing, data normalization, and statistical analysis" have been included in the revised manuscript. However, these details are not clearly found. Please ensure all such methodological details are explicitly presented in the manuscript.
7. The response letter mentions that "the data processing pipeline and analysis code have been made publicly available through GitHub: <https://github.com/mhorlbeck/ScreenProcessing>." However, this code appears to be from a previously published paper and not uniquely associated with the current study. Please provide access to the specific analysis scripts and pipeline used in this work.
8. Key information about the screening experiment is still missing, including PCR primer sequences used for library amplification, as well as sequencing parameters (e.g., single-end or paired-end reads, read length, sequencing depth). Please include these technical details in the Methods section.
9. Most importantly, for transparency and reproducibility, the authors should provide raw and processed data, as well as relevant scripts necessary to reproduce Figure 2b. This should include: (a) fastq files for the sgRNA NGS, (b) the list of sgRNA sequences in the library, and (c) sgRNA-level and gene-level output from the ScreenProcessing pipeline used for analysis.

Reviewer #3

(Remarks to the Author)

In extension to my previous comments on their first submission I would like to acknowledge, that the authors successfully improved the manuscript in its revised version. Additional experiments have been performed yielding supportive new data while previously submitted data is now omitted. The manuscript has been reorganized and streamlined, measures that improved the readability and flow of argumentation. Previous overstatements have been toned down.

However, during the revision process a new claim was introduced (e.g. to the 'Abstract') that is currently not sufficiently supported by data. The authors' attribution of the observed toxic effects to one principal lipid species appears as overinterpretation (and as unnecessary, maybe even conflicting the title – see my comments below.)

"Further mechanistic studies showed that m18:0/24:1 induces apoptosis through the mitochondrial permeability transition pore (mPTP) formation" in the 'Abstract' appears also overstretched. Experimental data from an in vitro swelling assay do not document apoptosis, a cellular process. Several papers cited by the authors suggest non-apoptotic processes as the underlying mechanism. Finally, the underlying mechanism may differ from cell type to cell type, an aspect not discussed by the authors. The in cellulo data of Fig. 5f only establish a link between 1-deoxySL toxicity (all, not a specific species) and mPTP. That link – in cells – with only one cell type investigated – may not be as direct as claimed by the authors extrapolating the in vitro data from the swelling assay.

Hence, if no additional data supporting above claims are presented the relevant text passages should be modified.

Additional points need to be addressed:

All abbreviations in the 'Abstract' need to be spelled out.

If claiming that Fig.1e shows "significantly higher cytotoxicity" please provide statistics.

Fig. 2b: Where would CerS5 and CerS6 be located in the volcano plot (see my original question L98)?

Fig. 2b,d: When analyzing the toxicity of deoxySphingolipids: If the knockdown of a gene conferred a significant resistance to 1-deoxySa induced toxicity should this gene really be termed "resistance gene" and its pathway "resistance pathway"? Along this, is FADS3 a "sensitizing gene" or a "de-sensitizing gene"? This nomenclature turns the cellular effect upside down.

If claiming that Fig.3d,h displays a "significant reduction" or significant protection" please provide statistics. Update legend to Fig.3 or provide IC50 values.

If claiming that Fig.4c demonstrates "significant protection" please provide statistics.

The statement "These results demonstrate that (22) is metabolically active in the nervous system..." should be used with caution (here and later in the 'Discussion'). Although the EFFECT is detected in the nervous system, the changed deoxy-SL profile here may result from the inhibitors' metabolic action outside the nervous system and e.g. lipid trafficking between organs. Has it been shown that the avian blood-brain-barrier is fully established at this developmental stage?

Omit "Comparable neuroprotective effects were observed in differentiated SH-SY5Y neuroblastoma cells, where (22) prevented neurite collapse following 1-deoxySa exposure (data not shown)." or show the data.

From the sentence "Lipidomics profiling confirmed the selective increase in the corresponding LC or VLC 1-deoxyDHCer species following supplementation (Figure 5b)" either "selective" or "LC" should be omitted.

Fig. 5b,c and also in the discussion: The authors state that (22) was added in these experiments to avoid further metabolism of the added FAs by ELOVL1. Yet, Fig.3a states that ELOVL1 elongates 20:1, 22:1 and 22:0, which were not added/assayed. A reader might not understand at once why (22) was added at all. In the rebuttal the authors explain this properly but have not added such explanation to the text. This should be corrected – at least with a few sentences and reference(s), if no additional figure is possible/needed.

The legend to Fig. 5b indicates a "vehicle control – BSA alone". If this is the green bars, please label as such. If not, please clarify.

If claiming that Fig.5d shows "most significant swelling response" please provide statistics. How does the bar look for the positive control (50uM CA2+)?

In the rebuttal, the authors state: "Our data showed that 1-deoxyDHCer (m18:0/24:1) induces mitochondrial swelling which is rescued by ... BAX inhibition...". Has this experiment been performed?

The second sentence of the 'Discussion' puts forward a strong message by narrowing it down to a single species. This

narrow statement is also repeated in the 'Abstract' (see my comment above). However, the authors show in their swelling assay that m18:0/24:0 shows ~60% and even m18:0/16:0 displays ~50% of the maximum toxicity. If this narrow statement is to be kept at both instances, it would need further support, e.g. analogous experiments as shown in Fig. 5e using m18:0/24:0, m18:0/16:0 and also analogous experiments as shown in Fig.5f using 1-deoxySA + (22) +/- CSA and/or BAXi.

The authors removed some data during the revision. Hence, "mitochondrial fragmentation" should be removed from the 'Discussion', too.

Please provide the source for the individual 1-deoxyDHCer, 1-deoxyCer, or canonical dihydroceramide (DHCer) species used in Fig. 5d.

All abbreviations should be harmonized e.g. Ceramide synthase is abbreviated as CerS and CERS.

The use of "knockout" and "knockdown" at several instances in the text needs to be doublechecked.

In the 'Author contributions' some initials remain flipped, e.g. MA <> AM or EY <> YE ...

Version 2:

Reviewer comments:

Reviewer #2

(Remarks to the Author)

The authors have made corrections and clarifications to some of my previous comments. However, some points remained to be addressed:

1. The information and citation of the CRISPRi v2 library used in this study is missing. Is it a commercial library or a custom library? Please provide the list of targeted genes and the corresponding sgRNA sequences included in the library.
2. The screen was CRISPRi-based, which does not cut DNA. However, in the diagram in Figure 2a a scissor icon was used to represent CRISPR cutting, which does not accurately reflect the screening platform. Please replace or correct this icon/label to accurately depict CRISPR interference (CRISPRi) rather than nuclease-mediated cutting.
3. The authors stated that the raw data associated with the screen are not available. Since the screen results are a key dataset for the paper, the absence of original/raw data prevents evaluation and reproduction of the findings. If the authors cannot provide the raw data, they need either repeat the screen and supply the raw data, or remove the screen results from the current manuscript.
4. The processed data are also missing in the current revision. These should include, at minimum: sgRNA count tables, the output files from the ScreenProcessing pipeline, and the dataset used to generate Figure 2b. Please submit these datasets as supplementary tables in the revision.

Reviewer #3

(Remarks to the Author)

The authors have addressed the raised concerns.

Point by Point Response

Reviewer #1 (Remarks to the Author):

This study provides a significant advance on our understanding of how toxic deoxysphingolipid species linked to diabetic polyneuropathy, MacTel, and HSAN1 mediate toxicity in disease. The authors employ an impressive combination of screens and assays to tease apart the specifics of a very complicated pathway. The CRISPRi screen and follow up analysis of the sphingolipid sub-species were very convincing. I have a couple comments regarding the mitochondrial toxicity assays and the hypothesis that ELOVL1/CERS2 co-expression explains cell specificity that I detail below. Overall this manuscript was well done and should be considered with revisions.

Specific comments:

1. For the silencing of CERSs and ELOVL1 using RNAi I did not see any data showing the specificity of the knock downs. Since both gene families have multiple genes with homologous sequences it would be appropriate to know the amount, if any, of crossover effect of each RNAi set on genes within the respective families. Some crossover is fine but we need to know the amount to make strong conclusions.

We agree this is an important point regarding RNAi specificity. We confirmed the specificity of the siRNA-mediated knockdown by RT-qPCR. The data are included in Figure 3b. Silencing of ELOVL1 did not alter the expression levels of other ELOVL isoforms, indicating minimal off-target effects within this gene family.

In addition, we supported the RNAi results with a new CRISPR-Cas9-generated ELOVL1 knockout model. Using ELOVL1, CerS2 and CerS5/6 double knockout HeLa lines, we independently recapitulated the observed lipidomics and phenotypic effects (Figures 3c–h). The complementary genetic approaches further strengthen our conclusion that ELOVL1 and CerS2 specifically mediate the formation of toxic 1-deoxyDHCer species.

2. It is unclear from the Seahorse experiments whether the effects on OCR and ECAR are due to direct effects on mitochondrial function or because the cells are dying and are undergoing apoptosis. The seahorse experiments are performed with 0.5uM lipid at 3 days which is when the authors can see up to 50% loss in cells in their survival assays. This seems to be consistent with how dramatic the effect of 1-deoxySa on background levels of OCR/ECAR following the rotenone/antimycin A injection in extended figure 1A, B. A complete loss of background ECAR would suggest that the cells are just dying and that's why you're seeing a loss of function. Can the authors find a change in mitochondrial function prior to death, either with a lower dose of 1-deoxySa or at an earlier time point before cell loss?

To ensure that our Seahorse data reflect mitochondrial dysfunction rather than nonspecific effects of cell death, all measurements were normalized to crystal violet (CV) staining, a viability marker applied after Seahorse analysis. This normalization accounts for cell loss and ensures that OCR and ECAR values reflect function per viable cell. We now indicate this normalization step in both the main text and figure legends (Figure 4d–f).

Furthermore, we performed mitochondrial swelling assays using freshly isolated mitochondria to independently confirm the direct effects of that 1-deoxySL on mitochondrial function,. The direct addition of purified 1-deoxyDHCer species, especially nervonyl-1-deoxyDHCer (m18:0/24:1), induced robust mitochondrial swelling (Figure 5d), supporting a direct mitotoxic effect. Importantly, this effect was prevented by co-treatment with cyclosporin A (CsA), a potent inhibitor of the mitochondrial permeability transition pore (mPTP) (Figure 5e), further indicating that 1-deoxySLs act on mitochondria directly.

These findings are in agreement with previously published studies demonstrating accumulation of 1-deoxySLs within mitochondria and their detrimental effects on mitochondrial structure and function (Alecú et al., *J Lipid Res* 2017; Gai et al., *Liver Int* 2020). Moreover, mitochondrial localization of 1-deoxySLs has been shown by imaging-based approaches, further supporting their direct presence in the organelle (Alecú et al., 2017; Lamberz et al., *bioRxiv* 2021). Together, these data support a model in which 1-deoxySLs—particularly 1-deoxyDHCer (m18:0/24:1)—compromise mitochondrial integrity through mPTP activation, leading to cell death.

3. The hypothesis that ELOVL1/CERS2 co-expression explains cell specificity is very interesting. If this were true, could you test it by looking for cell specific death in your DRG cultures following 1-deoxySa treatment?

While direct assessment of cell death within DRG cultures would require additional tools beyond the current study, we have strengthened our mechanistic understanding by adding new experimental data showing that 1-deoxySL-induced toxicity is mediated via BAX activation and mPTP opening. Using a BAX inhibitor peptide (V5) or cyclosporin A (CsA), a potent inhibitor of the mitochondrial permeability transition pore (mPTP), significantly rescued 1-deoxySa-treated cells from death (Figure 5). These findings indicate that BAX-dependent mitochondrial apoptosis is a key downstream consequence of toxic 1-deoxySL levels.

This mechanism is further supported by independent findings from Rosarda et al. (*Nat Commun*, 2023), who observed increased TUNEL staining in retinal organoids treated with 1-deoxySa (1 μ M), consistent with apoptosis in disease-relevant neural tissues.

While the hypothesis that co-expression of ELOVL1 and CerS2 underlies the specific neurotoxicity is compelling, experimentally testing this *in vivo* would require targeted genetic models, systemic 1-deoxySa delivery, and single-cell transcriptomic resolution—all of which pose substantial technical challenges given the high cytotoxicity of 1-deoxySLs.

We therefore consider this an important but rather longer-term aspect, which is beyond the scope of the present study. However, a certain specificity for neuronal tissue can already be associated from our present results as the Nervonic acid which converts in the most toxic 1-deoxySL (m18:1;24:1) is also most abundant in neuronal tissues. This is now also discussed in the text.

Reviewer #2 (Remarks to the Author):

This study investigates the mechanisms underlying the toxicity of 1-deoxysphingolipids (1-deoxySL), atypical sphingolipids associated with sensory neuropathies. Through a CRISPRi

toxicity screen in K562 cells, the researchers identified ELOVL1 and CerS2 as key factors mediating 1-deoxySL toxicity. Pharmacological inhibition of ELOVL1 prevented 1-deoxySL-mediated cellular and neuronal toxicity in DRG cells and SY5Y cells. The study also demonstrates that 1-deoxySL toxicity is linked to mitochondrial dysfunction and depends on the length and saturation state of both the long-chain base and the conjugated N-acyl chain, with Nervonyl-1-deoxyDHCeramide identified as the main mediator of toxicity.

Major points:

The toxicity phenotype of 1-deoxySL was established in HeLa cells (Fig. 1), but the screen was performed in K562 cells. To justify the choice of cell line, dose, and treatment duration for the screen, a similar toxicity phenotype should be demonstrated in K562 cells.

We are grateful to the reviewer for highlighting this point. Prior to initiating the CRISPRi screen, we optimized treatment conditions in K562 cells and confirmed that 1-deoxySa induces a dose-dependent cytotoxic phenotype in this model. Although the IC₅₀ in K562 cells was higher than that observed in HeLa cells, this is attributable to the different experimental parameters used for screening—such as plate format, cell density, media composition, and extended culture duration—which are not directly comparable to the conditions used in standard 3-day viability assays. Based on this optimization, we selected a concentration of 1.5 μ M 1-deoxySa, which approximates the IC₅₀ under screening conditions, as noted in the revised Methods section, Results, and Figure 2 legend.

To further support the robustness of our findings, we have also tested 1-deoxySa toxicity across eight additional human cancer cell lines (Caco-2, DLD1, HT29, HCT116, SW480, SW620, SW1116, and T84), where IC₅₀ values ranged from 0.2 to 0.5 μ M, consistent with HeLa cells under experimental conditions described in the manuscript. These results demonstrate that the toxic effects of 1-deoxySLs are broadly reproducible across diverse cell types and responsive to ELOVL1 inhibition. These additional data will be included in a follow-up study.

We have revised the manuscript accordingly to clarify the rationale for using K562 cells in the screen and to highlight the broader reproducibility of the observed toxicity phenotype.

The authors suggest that the co-expression of ELOVL1 and CerS2 in specific DRG cell subtypes may explain the selective toxicity of 1-deoxySA in HSAN1 patients. To support this hypothesis, in vivo toxicity experiments combined with single-cell analysis would be beneficial.

Thanks for this suggestion. The hypothesis is based on data presented in Supplementary Table 1, which were extracted and adapted from a dataset co-submitted with the referenced Nature Neuroscience study. The study categorized single-cell RNA sequencing data of murine DRG neurons. These data indicated a co-expression of ELOVL1 and CerS2 in specific nociceptor subtypes, supporting our model that metabolic co-expression may underlie the selective vulnerability of sensory neurons to 1-deoxySL toxicity.

However, confirming this hypothesis in vivo would require to repeat this complex and technically demanding study. Specifically, it would involve:

1. Performing single-cell transcriptomics on mouse DRGs after systemic 1-deoxySa exposure, and
2. Assessing whether co-expressing subpopulations exhibit selective loss or altered gene expression profiles.

Such an approach would likely require a genetic interference or lineage tracing strategy in combination with an in vivo HSAN1 model. Unfortunately, none of the currently available HSAN1 mouse models recapitulates the neurological phenotype. In addition a systemic supplementation of 1-deoxySL in rodent model has not been established yet. The cytotoxicity of 1-deoxySa presents challenges for systemic dosing without off-target toxicity. Moreover, single-cell RNA sequencing post-treatment is unlikely to recapitulate the same clustering and resolution observed under physiological conditions, as dying or damaged neurons may be underrepresented or transcriptionally unstable.

We believe this is an important future direction, but these experiments are outside the scope of the current study. We have adapted the text and discuss these limitations and the rationale for our hypothesis in the revised discussion.

The description of the screen methodology and analysis is incomplete and unclear. Essential information is missing, including number of cells used, library coverage, treatment dose and duration, number of replicates, raw sequencing data and processed screening results.

We appreciate the reviewer's comment and have now thoroughly revised the Methods section to include all relevant experimental details related to the CRISPRi screen. Specifically, we now provide the following information:

1. The number of cells transduced and maintained throughout the screen
2. The library coverage and multiplicity of infection
3. The concentration of 1-deoxySa used (1.5 μ M), selected based on pre-screen IC50 determination under screen-specific conditions
4. The duration of treatment (5 cell passages),
5. The number of biological replicates
6. Details regarding sequencing, data normalization, and statistical analysis

To promote transparency and reproducibility, the data processing pipeline and analysis code have been made publicly available through GitHub: <https://github.com/mhorlbeck/ScreenProcessing>

Additionally, the processed screening results will be provided as supplementary materials, and Figure 2 legend have been updated accordingly.

We thank the reviewer for helping us improve the clarity and completeness of this section of the manuscript.

Crucially, the screen was CRISPRi-based, which inhibits gene expression at the transcriptional level rather than introducing mutations. However, the analysis described in the methods section is based on determining mutation frequencies, which is inappropriate for CRISPRi. Please clarify the analysis method used and ensure it aligns with the CRISPRi approach.

Due to a miscommunication, the original description in the Methods section inaccurately referenced mutation frequencies, which is indeed not applicable to CRISPRi-based screens. This has now been corrected and revised to accurately reflecting the analytical workflow used. The updated Methods section now describes the CRISPRi-compatible analytical pipeline, and we have also provided a link to the publicly available code used for data processing: <https://github.com/mhorlbeck/ScreenProcessing>

Minor points:

Scale bars are missing for Figure 4E.

Font sizes in figures are inconsistent and sometimes too small to read. Please standardize font sizes throughout all figures for improved readability.

We have carefully revised and standardized font sizes across all figures to ensure consistency and readability. All text, axis labels, and legends have been adjusted to meet publication-quality standards, and updated figure versions have been included in the revised manuscript.

Reviewer #3 (Remarks to the Author):

In the manuscript by Majcher et al. the authors demonstrate a so far unacknowledged role of ELOVL1 and CerS2 for 1-deoxySL toxicity. Both enzymes are involved upstream of the 1-deoxyDHCer synthesis. They contribute one building block of 1-deoxyDHCer, the VLC-FA, yielding ultimately VLC-1-deoxyDHCer, if deoxy-LCB are also in supply. While it has been discussed before, that the N-acylated 1-deoxySL might mediate the toxicity of 1-deoxySL, no published data directly shows an involvement of ELOVL1 and CerS2. As both enzymes have a long-established function in FA elongation and ceramide synthesis, their role for VLC-1-deoxySL synthesis might have been expected. The finding that the VLC-members are primarily responsible for the toxicity of 1-deoxySL is intriguing and at the heart of this manuscript. It opens up the possibility to influence the detrimental process in affected patients.

The methodology applied in the study is sound. The work meets the expected standards in the field.

In its current form the manuscript shows formal shortcomings. It appears hastily assembled from a previous submission elsewhere, without final proofing of the new version. An Extended Figure 1 is available, but not called in the text; Extended Figures 2 and 3 are not existing; Extended Figure 4 is called but not available. Confusing. A revision, including further experiments is needed.

We thank the reviewer for their detailed and constructive feedback.

We sincerely apologize for the formatting inconsistencies in the previous version and acknowledged that some internal references to extended figures were incomplete and misleading.

We now carefully reviewed and corrected all figure references throughout the manuscript, figure legends, and main text. Extended Figure 1 is now properly cited, and Extended Figures 2 and 3 have been removed to avoid confusion. The structure of the figures has been adopted to the comments in order ensure better readability of the manuscript.

In addition to these formal aspects, we have expanded the dataset with new experiments that significantly strengthen our core findings.

Beside a newly develop live chicken embryo model described below, we identified a new mechanistic link between VLC-1-deoxyDHCer toxicity and mitochondrial dysfunction. Our data showed that 1-deoxyDHCer (m18:0/24:1) induces mitochondrial swelling, which is rescued by cyclosporin A (mitochondrial permeability transition pore inhibitor, mPTP inhibitor) and BAX inhibition, establishing BAX- and mPTP-mediated cell death as a novel mechanism of 1-deoxySL cytotoxicity.

We believe these revisions and additions substantively enhance the manuscript and more clearly support our central hypothesis regarding the structure–function relationship of 1-deoxySLs and the role of VLC acyl chain in mediating toxicity. We are grateful for the reviewer’s observations, which helped us substantially improve the quality, clarity, and completeness of our submission.

Importantly, data on a more complex disease model such as HSAN1 or diabetic mice would considerably strengthen the manuscript and appear necessary for publication in this journal.

We fully agree that validation of our findings in a more complex in vivo disease model would strengthen the translational relevance of the study. Unfortunately, this is currently not well feasible due to limitations in available animal models. Previously published models either show only mild elevations in 1-deoxySLs or fail to develop a neuropathy phenotype, likely due to species-specific differences in sphingolipid metabolism (Hines, T. J. et al. *Precision mouse models of Yars/dominant intermediate Charcot-Marie-Tooth disease type C and Sptlc1/hereditary sensory and autonomic neuropathy type 1*. J Anat (2021) and <https://www.jax.org/strain/032194>)

A previously reported HSAN1 model which systemically expressed the mutant under a strong promotor is not available anymore. Importantly, the sphingolipidome in mice is significantly distinct to humans with inherently lower SL levels. Rodents also have a larger set of CYP4F enzymes, which are converting 1-deoxySLs into potentially less toxic downstream derivatives, which might also explain their resistance to the toxic effects compared to humans.

However, to address this limitation and provide further supportive data from a more complex system, we have now included a chicken embryo in vivo flux model. In this model. Developing chicken embryos were systemically treated with isotopically labeled d₃-1-deoxySa, and DRGs were isolated for high-resolution LC-MS/MS analysis. These data confirm that 1-deoxySLs are absorbed and metabolized in DRG neurons in vivo. Importantly, co-treatment with the ELOVL1 inhibitor compound 22 significantly reduced levels of the most toxic species, nervonyl-1-deoxyDHCer (m18:0/24:1) (Extended Figure 4a,b). However, in-vivo neurotoxicity was not addressed in the model out of technical reasons.

Additionally, in dissociated DRG-derived sensory neurons, we demonstrated that ELOVL1 inhibition by compound 22 effectively prevented 1-deoxySL-induced neurite loss, further supporting its neuroprotective potential (Extended Figure 4d,e).

While we agree that testing compound 22 in a mammalian disease model would be of high interest, it is important to note that compound 22 was originally developed for X-ALD and specifically optimized to cross the BBB to inhibit ELOVL1 in the CNS. All currently available ELOVL1 inhibitors share this limitation. Therefore, future progress in this area will require both the development of peripherally restricted ELOVL1 inhibitors and the generation of improved in vivo neuropathy models. We recognize both aspects as a priority, which however, is currently beyond the scope of this study.

In addition, we are currently in discussions with industrial partners to explore the translational potential of our findings. This manuscript establishes a previously unrecognized pathomechanism linking the structure of individual 1-deoxySL species to BAX- and mPTP-mediated mitochondrial cell death, providing a strong mechanistic foundation for future therapeutic development.

The ‘Abstracts’ suffers from overstatements and should be toned down, e.g. in line26: While the manuscript establishes a role for CerS2 and ELOVL1, their role as ‘key enzymes’ is currently not fully substantiated. Please omit “the key enzymes” from this sentence.

We agree that the original wording in the abstract is too strong and may have implied a level of exclusivity that is not yet fully supported. While our data—supported by both an unbiased CRISPRi toxicity screen and functional validation experiments—demonstrate that CerS2 and ELOVL1 play a critical role in mediating 1-deoxySL toxicity, we acknowledge that other enzymes or regulatory factors may also contribute to this process.

We have toned down the language in the abstract and also removed “key enzymes.” The revised text now reflects the contribution of CerS2 and ELOVL1 without overstating their exclusivity or hierarchical role.

Also, can the authors substantiate the claim in line31 that “this toxicity depends on the length ... of the long-chain base”?

Our intention was to refer to the N-acyl chain length of the fatty acid conjugated to the 1-deoxysphingoid base, not the long-chain base itself. This was a misstatement in the original text. We have now corrected the sentence throughout the manuscript to more accurately reflect that toxicity is dependent on the length and saturation of the N-acyl chain, rather than the sphingoid base.

We appreciate the reviewer’s attention to this point, which helped us improve the precision of our interpretation and wording.

Other points include:

L61: has been

L72: bad call of figure - Fig. 1A does not match the meaning of this sentence

L79: bad call of figure - Fig. 1A does not match the meaning of this sentence

The text and figure references have been corrected in the revised manuscript.

Fig. 1B: How do these curves look for HeLa cells overexpressing FADS3?

It has been previously demonstrated that HeLa cells endogenously express high levels of FADS3, as reported by Karsai et al. (J Biol Chem, 2020), where also a high 1-deoxyCer/1-deoxyDHCer ratio indicated active $\Delta 14Z$ desaturation. In agreement with this, we observed a similar metabolic profile in our HeLa model, with a 1-deoxyCer/1-deoxyDHCer ratio of approximately 4:1 (Figure 1 g). In the same study, overexpression of FADS3 in HEK293 cells, which have low baseline FADS3 activity and expression, significantly reduced the cytotoxic effects of 1-deoxySa, while siRNA-mediated FADS3 knockdown in HeLa cells resulted in increased toxicity. These findings support the hypothesis that FADS3 has a protective effect by converting toxic 1-deoxyDHCer to the less toxic 1-deoxyCer.

In the current study, we adopted a complementary approach to functionally distinguish the toxic species by comparing the effects of 1-deoxySa (m18:0), which can be metabolized to 1-deoxyDHCer AND 1-deoxyCer while 1-deoxySo (m18:1) is exclusively converted to 1-deoxyCer. In combination with FB1, which blocks N-acylation, this strategy enabled us to attribute the cytotoxic phenotype specifically to 1-deoxyDHCer species.

However, to address the reviewer's question, we additionally performed FADS3 overexpression experiments in HeLa cells using a pcDNA-FADS3 construct. This resulted in a reproducible and statistically significant increase in the IC50 for 1-deoxySa toxicity from 0.18 μM to 0.26 μM , indicating a moderate but measurable protective effect. We observed a comparable IC50 shift in FADS3 overexpressing HCT116 cells, further validating this observation across distinct cellular contexts.

Importantly, FADS3 was not identified as a significant hit in our genome-wide CRISPRi toxicity screen, suggesting that although it contributes to modulating 1-deoxySL toxicity, its impact is less compared to ELOVL1 and CERS2, which emerged as the top genetic hits in our study.

L86: introduce line break

was corrected

L88: No information provided on the concentration of 1-deoxySA during screening. How does the 1-deoxySA susceptibility of K562 cells compare to HeLa?

Prior to initiating the CRISPRi screen, we performed toxicity assays in K562 cells to determine the appropriate screening concentration of 1-deoxySa. These experiments demonstrated that K562 cells are less sensitive to 1-deoxySa than HeLa cells, with a higher IC₅₀ value. However, this difference must be interpreted in the context of distinct experimental conditions used for the screen—such as plate format, seeding density, media composition, and prolonged culture duration—which are not directly comparable to the short-term viability assays used in HeLa cells.

Based on this optimization, we selected a 1-deoxySa concentration of 1.5 μ M for the CRISPRi screen, which approximates the IC₅₀ in K562 cells under the specific screening conditions. This concentration was used consistently across all replicates to allow sufficient selective pressure while preserving cell viability and library coverage. This concentration, as well as the rationale, has now been included in the Results, Methods, and Figure 2 legend.

To further support the reproducibility and broader relevance of our findings, we also evaluated 1-deoxySa toxicity in eight additional human cancer cell lines (Caco-2, DLD1, HT29, HCT116, SW480, SW620, SW1116, and T84). Across all tested lines, we observed IC₅₀ values ranging from 0.2 to 0.5 μ M under standardized toxicity assay conditions. Importantly, in all cases, co-treatment with the ELOVL1 inhibitor compound 22 rescued the cytotoxic phenotype, demonstrating that the ELOVL1-dependent mechanism is conserved across multiple cell types. These additional findings further validate the central conclusions of our study and are part of a manuscript currently in preparation focused on therapeutic targeting of ELOVL1 in cancer contexts.

L95: check grammar

Grammar was adjusted.

L98: Where would CerS5 and CerS6 be located?

All Ceramide Synthase (CERS) isoforms, including CERS5 and CerS6, are localized to the endoplasmic reticulum (ER), as broadly reviewed by Mullen et al. (Biochem J, 2012). Specific ER localization of CERS5 was demonstrated by Laviad et al. (J Biol Chem, 2012), and CERS6 ER colocalization has been shown in studies such as Lee et al. (J Biol Chem, 2022). We have now revised the manuscript text.

L100/1/2: biosynthesis or the elongation; ribosomes; oxidative

L134: omit “a”

L136: no line break

L141: m18:0/18:0 does not appear to be changed

We corrected the text.

L174: Extended Figure 4 is not available – you may not call Extended Figure 4 before calling Extended Figures 1-3 in the text

We apologize for the mistake and we corrected the text.

L170+175: Are refs 19 and 27 the same?

Yes, thank you for pointing this out. The references have been reviewed and corrected accordingly.

L179: Extended Figure 4 is not available – you may not call Extended Figure 4 before calling Extended Figures 1-3 in the text

Figure references were adapted and the whole manuscript revised and corrected.

Fig. 4D: What does one data point represent? One cell, one analyzed image, ...

In Figure 4D (now Extended Figure 4 d in the revised manuscript), each data point represents the total neurite network area quantified from an individual well. For each well, the entire surface was imaged using automated tiling and stitching, and the total neurite area was computed through quantitative image analysis of the full well image. This approach captures global neurite outgrowth rather than relying on selected fields of view, and ensures a more representative and reproducible measurement per condition.

Each condition included multiple biological replicates, with one data point per replicate well, as indicated in the figure legend. We have clarified this point in the revised manuscript and updated the legend of Extended Figure 4 d to reflect the exact nature of the quantification.

Fig. 4E: Panels e-l need to be presented as greyscale images. The TOMM20 signal appears to be diffuse and mitochondria are hard to discern – is this an image compression artifact? Please make sure the arrows point directly(!) to mitochondria. Show larger enlargements.

We have updated Figure 4E to improve image clarity and interpretability.

Neurofilament staining is now presented in greyscale, as recommended, to enhance contrast and structural definition of neurites.

We also removed the TOMM20 signal from this panel. Due to the limited number of surviving neurons in the 1-deoxySa-treated condition, mitochondrial signal quality was insufficient to draw robust conclusions at the image level.

Instead, we have focused on more quantitative and functional assessments of mitochondrial dysfunction, which are now clearly presented in the Seahorse assay data (Figure 4 d, e, f). Additionally, we provide direct evidence of mitochondrial damage through swelling assays in isolated liver mitochondria upon treatment with 1-deoxyDHCer species (Figure 5 d, e), supporting our mechanistic hypothesis.

Together, these quantitative functional assays provide additional mechanistic evidence of mitochondrial involvement than the TOMM20 immunofluorescence data that were shown in the previous version. These changes have been incorporated in the revised manuscript and corresponding figure legends. We appreciate the reviewer's input, which helped us improve both the clarity and scientific rigor of this section.

L191: “significantly” - What quantification and statistical analysis are you referring to?

The figure legends were revised.

L193/5: Extended Figure 4 is not available – you may not call Extended Figure 4 before calling Extended Figures 1-3 in the text; Does this manuscript contain data on SH-SY5Y cells? If not, please revise also M&M.

We have revised the manuscript to ensure that Extended Figures are cited in the correct order. The numbering and references have been corrected and now appear sequentially in the text.

Regarding the SH-SY5Y cells, we had initially included qualitative data on neurite protection by compound 22 in differentiated SH-SY5Y cells, which were consistent with our DRG data. However, since these results were not quantified and were not essential to the central conclusions of the study, we have removed all SH-SY5Y data from the manuscript, including the Methods section.

L204: and glycolytic capacity

L213: The mentioned scale bars are missing in Fig. 4E.

The figure was adapted.

L238: Where in M&M can this protocol be found? How is the solubility of ceramides secured?

The mitochondrial swelling assay protocol has now been clearly described in the revised Methods section.

Specifically, 1-deoxyDHCer, 1-deoxyCer, and canonical DHCer were dissolved in ethanol, and small volumes of these ethanolic solutions were directly added to isolated mitochondria, consistent with protocols previously published by Novgorodov et al. (J Biol Chem, 2008). The final ethanol concentration did not exceed 0.5% (v/v) in any condition and vehicle controls were included accordingly.

We have added the appropriate reference and describe the procedure in the Methods section for transparency and reproducibility.

Fig. 5A: doxicity > toxicity

We actually thought about the term “Doxicity” as a wordplay due to its analogy to “toxicity”. But we agree that this is not compliant with the required scientific seriousness, We therefore revised the figure legend and replaced the term with “toxicity” for clarity and consistency throughout the manuscript

Fig. 5B,C: How would these panels look in the absence of inhibitor 22? How in the presence of the pan-CerS-inhibitor FB1 instead of inhibitor 22?

This is an excellent question. The experimental design for Figure 5 b, c was specifically designed to address the toxicity of individual VLC 1-deoxyDHCer species. The approach allowed us to selectively modulate the acyl chain composition of 1-deoxyDHCer species and link these profiles functionally to toxicity. The inhibition of ELOVL1 prevented the further elongation of endogenous fatty acids, which might bias the results.

Performing the same assay in the absence of ELOVL1 inhibitor 22 would result in uncontrolled elongation of supplemented fatty acids, in particular for the LC-FAs (e.g., C16:0) leading to

mixed metabolic backgrounds and overlapping 1-deoxySL species. We already demonstrated this earlier using UC13-palmitate (M+16) tracing in HeLa cells coupled with high resolution LC-MS/MS lipidomics analysis, which revealed incorporation into both long- and very-long-chain sphingolipids (via M+16 and M+32 species), confirming that palmitate undergoes elongation under the assay conditions.

Additionally, labeling studies in sterol esters confirmed that elongation of 16C13-palmitate occurs independently of the sphingolipid pathway, highlighting global VLCFA generation in the presence of active ELOVL1.

In addition, CerS2 activity in HeLa cells is high, which lead to a bias towards the formation of VLC-Cer (assay adapted from Couttas et al., Lipids, 2015), .

Therefore, without ELOVL1 inhibition, supplemented LC-FAs would contribute indirectly to VLC-1-deoxySL formation, confounding interpretation.

FB1 is known to be acylated by CerS enzymes, forming acyl-FB1 species, which have been reported to be cytotoxic as well (Zhang et al., Structure, 2024). Supplementation with free fatty acids in the presence of FB1 would likely enhance intracellular acyl-FB1 formation, creating a complex and undefined mixture of toxic species (acyl-FB1, 1-deoxySLs, canonical SLs, and LCBs), which limits the interpretation of 1-deoxyDHCer toxicity.

atty acid supplementation in combination ELOVL1 inhibition is therefore the most robust framework for examining the structure–toxicity relationship of individual 1-deoxySL species. This approach was further validated by our mitochondrial swelling assay, where nervonyl-1-deoxyDHCer (m18:0/24:1) emerged again as the most toxic species.

Fig. 5D and L283: What exactly is the “indicated treatment”? No specifics given in M&M either. Please explain the assay and give reference (if applying): Why is there a decline in absorbance 540nm? What were the controls?

A mitochondrial swelling assay is a well-established method to monitor mitochondrial permeability transition pore (mPTP) opening, as opening induces osmotic swelling in isolated mitochondria. This swelling is measured by light scattering at 540 nm.

Figure 5D “indicated treatment” refers to the direct addition of purified 1-deoxyDHCer, 1-deoxyCer, and canonical dihydroceramide (DHCer) species to freshly isolated mouse liver mitochondria. Lipids were dissolved in ethanol and added to the mitochondrial suspension at final concentrations described in the revised Methods section. The vehicle control (ethanol alone) was used for normalization, and Ca²⁺ treatment (50 μM) served as a positive control for mPTP activation.

We have now clarified this in the Methods section, also specifying treatment conditions, measurement parameters, normalization strategy, and positive controls. Additionally, we have cited the relevant reference describing the methodological foundation

L267: add “in”

Text was adjusted.

L288: Move caption to the top of the table. Add reference to previously published data (if applying).

Text was adjusted.

Extended Figure 1 is not called in the text and appears a fragment. Is this Extended Figure 4? *If so, the fragmentation of mitochondria in panel C cannot be recognized (potentially due to image compression). Please provide suitable images and some sort of quantitation of the phenomena. The desired information in panel D is lost altogether (potentially due to image compression). Please provide suitable images of the phenomena.*

To avoid misinterpretation and improve clarity, we removed the TOMM20 mitochondrial staining images from the extended figure. We decided that the image resolution and signal intensity were insufficient for a reliable segmentation and quantitative analysis of mitochondrial morphology under the 1-deoxySa treatment condition.

To assess change in mitochondria on a functional level, we employed quantitative assays, including Seahorse XF analysis (Figure 4 d, e, f) and mitochondrial swelling assays using isolated mitochondria (Figure 5 d, e), both of which support a direct impact of 1-deoxySLs on mitochondrial function. Figure numbering and in-text references have been corrected accordingly.

Discussion: Revise and sharpen. Avoid overly borrowing text passages from earlier parts of the manuscript. Please check journal style on calling figures in the ‘Discussion’.

The Discussion has been revised and refined to improve clarity, avoid repetition from earlier sections, and to better highlight key findings. We have also incorporated new data supporting our central hypothesis, and carefully adapted figure citations

L349/50: Where is this data to be found (see my comment on Fig. 5B,C above)?

L363: “Retraction from axons”- Do you show data on this? Is this different from the data on ‘neurite area’ in Fig. 4D?

Thanks for pointing that out. The relevant text and figure references have been revised for clarity and accuracy. We now also adjusted the description of axonal effects to avoid over-interpretation and ensured that it aligns with the quantified neurite area data. No separate evidence of axon retraction is provided in this paper. We updated the manuscript text accordingly to reflect this.

Please check journal style on capitalization: L-Alanine or L-alanine; Serine-palmitoyltransferase or serine-palmitoyltransferase etc. in 'Abstract' and elsewhere.

Please clarify author contribution: It appears puzzling that MA performed all(!) the experiments and analysis, interpreted the data and wrote the manuscript but is not an author, while the manuscript has two joint-first-authors, whose contribution is unclear.

The abbreviation "MA" refers to Adam Majcher, who is the first author of the manuscript and indeed performed and verified most of the experiments in the paper. The author list and contribution statement now reflects the individual roles and clarifies the designation of the joint first authorship.

REVIEWER COMMENTS

Reviewer #1 (Remarks to the Author):

The authors have addressed all of my concerns.

minor not there is a typo on page 17 "This demonstrates that 1-deoxyDHCer (m18:0/24:1) directly perturbs mitochondrial integrity." should read "integrity"

We thank the reviewer for his contribution which significantly improved the manuscript. We have corrected the typographical error on page 17.

Reviewer #2 (Remarks to the Author):

I still have the following points regarding the CRISPRi screen in the revised manuscript:

1. In the Results section, the authors state: "Haploid K562 cells expressing CRISPRi-dCas9-KRAB were transfected." However, K562 cells are generally considered diploid, albeit with chromosomal abnormalities. Please correct this statement or provide clarification regarding the haploid status of the K562 cells used.

Thanks for pointing this out. We acknowledge that K562 cells are not haploid and exhibit a complex karyotype with chromosomal abnormalities. To avoid misrepresentation, we have removed the term "haploid" from the manuscript. The text has been corrected accordingly.

2. In the legend for Figure 2b, the authors state: "the difference in sgRNA as quantified by RNA sequencing." However, sgRNA abundances in CRISPR screens are typically quantified by DNA sequencing, not RNA sequencing. Please revise this description to accurately reflect the methodology used.

We apologize for this error. The sgRNA abundances were of course quantified by DNA sequencing, not RNA sequencing. The figure legend and corresponding text have been revised to accurately reflect this methodology.

3. There is no legend provided for Figure 2d. Please add an appropriate figure legend for this panel.

Legend corrected.

4. In the Methods section, the authors mention: "10 computer-generated, non-targeting sgRNAs clustered into 15K 'pseudogenes'." It is unclear how 10 non-targeting sgRNAs could generate 15,000 "pseudogenes." Please provide a more detailed explanation regarding this approach.

We thank the reviewer for pointing this out. As described in Horlbeck et al. (eLife, 2016), a small number of non-targeting control sgRNAs were computationally assigned to a large set of artificial "pseudogenes" to establish a robust background distribution for statistical comparisons. Specifically, 10 non-targeting sgRNAs were randomly distributed and re-used to simulate the behavior of 15,000 pseudogenes, thereby approximating the null distribution and enabling gene-level significance scoring using tools such as MAGeCK or ScreenProcessing. This approach avoids inflation of multiple testing corrections while maintaining a realistic representation of variance within control sgRNAs.

5. The sentence: "In all, 350K sgRNAs were used with a 60% targeting, 40% non-targeting ratio." requires further clarification. Please elaborate on the composition of the sgRNA library and the rationale for this ratio.

As detailed in Horlbeck et al. (eLife, 2016), the CRISPRi library includes both gene-targeting and non-targeting sgRNAs. Approximately 60% of the sgRNAs are designed to target annotated genes, with five sgRNAs per transcription start site (TSS), selected based on predicted efficacy and specificity. The remaining 40% of the library comprises non-targeting sgRNAs and other control sequences, which serve to model the distribution of non-specific effects and allow accurate normalization and false discovery rate (FDR) estimation. This composition was empirically optimized to enhance screen robustness and statistical power.

6. The response letter states that information on “the number of biological replicates” and “details regarding sequencing, data normalization, and statistical analysis” have been included in the revised manuscript. However, these details are not clearly found. Please ensure all such methodological details are explicitly presented in the manuscript.

We thank the reviewer for the helpful comment. We now explicitly state in the revised Methods section that the CRISPRi screen was performed in two independent biological replicates per condition. Regarding data normalization and statistical analysis, these were performed using the publicly available ScreenProcessing pipeline (Horlbeck et al., eLife, 2016), which implements normalization across conditions, control-based modeling, and Mann-Whitney U tests for gene-level significance scoring. We have revised the manuscript to specify clearly the number of replicates and added a sentence summarizing the analysis pipeline, including a citation and GitHub link for transparency and reproducibility.

7. The response letter mentions that “the data processing pipeline and analysis code have been made publicly available through GitHub: <https://github.com/mhorlbeck/ScreenProcessing>”; However, this code appears to be from a previously published paper and not uniquely associated with the current study. Please provide access to the specific analysis scripts and pipeline used in this work.

We confirm that the referenced GitHub repository (<https://github.com/mhorlbeck/ScreenProcessing>) contains the exact data processing pipeline and analysis code used in our study. While originally developed and described in the Horlbeck et al. publication, this pipeline is designed for general application across CRISPRi screen datasets and is not restricted to any one study. All input files and parameters specific to our analysis are fully compatible with the repository's structure. We now revised the methods section.

8. Key information about the screening experiment is still missing, including PCR primer sequences used for library amplification, as well as sequencing parameters (e.g., single-end or paired-end reads, read length, sequencing depth). Please include these technical details in the Methods section.

Thank you for pointing this out. We have now included the PCR primer sequences used for sgRNA library amplification, as well as the sequencing parameters. These details were adapted from Horlbeck et al., 2016 (PMID: 27723780), which we followed closely in our experimental design. The methods section has been updated accordingly.

9. Most importantly, for transparency and reproducibility, the authors should provide raw and processed data, as well as relevant scripts necessary to reproduce Figure 2b. This should include: (a) fastq files for the sgRNA NGS, (b) the list of sgRNA sequences in the library, and (c) sgRNA-level and gene-level output from the ScreenProcessing pipeline used for analysis.

We fully agree with the importance of transparency and reproducibility in large-scale functional screens. Unfortunately, the original FASTQ files were stored on institutional servers at Washington University and are no longer accessible to us following institutional transition. Despite efforts to retrieve them during revision, we have been unable to recover the raw sequencing data. We are currently in contact with Washington University IT services to investigate whether data recovery remains possible, and we will make the raw sequencing data publicly available if retrieval is successful.

To support reproducibility of the findings in the meantime, we can provide the normalized \log_2 fold-change (\log_2FC) and $-\log_{10}(p\text{-value})$ statistics for the entire screen in supplementary.

We acknowledge the absence of the FASTQ files and appreciate the reviewer's understanding. We are committed to transparency and will continue working to recover and share the full raw dataset.

Reviewer #3 (Remarks to the Author):

In extension to my previous comments on their first submission I would like to acknowledge, that the authors successfully improved the manuscript in its revised version. Additional experiments have been performed yielding supportive new data while previously submitted data is now omitted. The manuscript has been reorganized and streamlined, measures that improved the readability and flow of argumentation. Previous overstatements have been toned down.

However, during the revision process a new claim was introduced (e.g. to the 'Abstract') that is currently not sufficiently supported by data. The authors' attribution of the observed toxic effects to one principal lipid species appears as overinterpretation (and as unnecessary, maybe even conflicting the title – see my comments below.)

“Further mechanistic studies showed that m18:0/24:1 induces apoptosis through the mitochondrial permeability transition pore (mPTP) formation” in the 'Abstract' appears also overstretched. Experimental data from an in vitro swelling assay do not document apoptosis, a cellular process. Several papers cited by the authors suggest non-apoptotic processes as the underlying mechanism. Finally, the underlying mechanism may differ from cell type to cell type, an aspect not discussed by the authors. The in cellulo data of Fig. 5f only establish a link between 1-deoxySL toxicity (all, not a specific species) and mPTP. That link – in cells – with only one cell type investigated – may not be as direct as claimed by the authors extrapolating the in vitro data from the swelling assay.

We thank the reviewer for this important observation. We agree that apoptosis, as a complex cellular process, cannot be definitively concluded from the mitochondrial swelling assay alone, and we have revised the wording in the Abstract and Discussion accordingly to avoid overstatement.

While our swelling assay directly demonstrates that 1-deoxyDHCer (m18:0/24:1) compromises mitochondrial integrity via mPTP activation, we agree this in vitro assay does not alone confirm apoptotic cell death. However, we complemented these findings with cell experiments (Fig. 5f), where co-treatment with the mPTP inhibitor CsA significantly rescued 1-deoxySL-induced toxicity. Although these experiments were conducted in a single cell type in this study, we have replicated these results in two additional cell lines—these experiments focused on the role of 1-deoxySLs in colon cancer and will be included in a later manuscript

We have now clarified this limitation in the text and reframed the conclusions to reflect the evidence more accurately, emphasizing that our findings point to a role of mPTP in mediating cytotoxicity, but do not fully define the mode of cell death as apoptosis. Additionally, we acknowledge that cell-type-specific differences may influence the underlying mechanisms and have included this limitation in the revised Discussion.

Hence, if no additional data supporting above claims are presented the relevant text passages should be modified.

Additional points need to be addressed:

All abbreviations in the 'Abstract' need to be spelled out.

Text double checked.

If claiming that Fig.1e shows “significantly higher cytotoxicity” please provide statistics.

We appreciate the reviewer’s emphasis on rigorous statistical measures. Indeed, IC50 and full dose–response curve fitting are recognized as robust and integral methods for comparing compound potencies in pharmacological and toxicological studies. As outlined by Srinivasan and Lloyd (2024), IC50 values derived from properly modeled dose–response curves allow for meaningful comparisons across compounds and are less susceptible to distortion from biological variability or non-linear response behaviour. (Srinivasan & Lloyd, *J. Med. Chem.* 2024, 67 (20), 17931–17934; <https://doi.org/10.1021/acs.jmedchem.4c02052>). Moreover, methodological guidance strongly recommends model-based analysis of dose–response data: a comprehensive review in *Archives of Toxicology* emphasizes that parametric modeling and alert-concentration metrics (such as IC50) are superior to ad hoc pointwise comparisons, especially for detecting and quantifying toxic effects. (Kappenberg *F Arch Toxicol.* 2023 Oct;97(10):2741-2761. doi: 10.1007/s00204-023-03561-w. Epub 2023 Aug 12. PMID: 37572131; PMCID: PMC10474994)

In our assays, 1-deoxySO did not reduce cell viability by more than 50% in a dose-dependent manner within the tested concentration range (0–3 μM), precluding the calculation of an IC50. In contrast, 1-deoxySA treatment showed a clear and reproducible dose–response effect, with IC50 values ranging from 0.394 to 0.482 μM across replicates. In addition, this stark difference in the cytotoxicity profiles provides strong evidence that 1-deoxySA is substantially more toxic than 1-deoxySO under identical experimental conditions and within a physiological relevant conc. range.

While pointwise statistical comparisons (e.g., at individual concentrations) can be informative, we intentionally focused on IC50 estimation as it captures the entire response curve and allows for a more integrative and statistically meaningful comparison of compound potency. Moreover, single-point comparisons may either overestimate or underestimate toxicity differences due to inherent biological variability or non-linear response dynamics. We have clarified this rationale in the revised figure legend and Results section to better support the interpretation of Figure 1.

Fig. 2b: Where would CerS5 and CerS6 be located in the volcano plot (see my original question L98)?

CerS5 was identified as statistically significant in the CRISPRi screen, but its \log_2 fold change (logFC) was relatively low, placing it outside the selected cutoff for visualization in the volcano plot. This likely reflects its limited contribution to 1-deoxySL toxicity compared to CerS2, which shows a stronger phenotype and is more specific for very-long-chain acyl-CoAs, as supported by our data and previous literature.

CerS6 was not significant

Importantly, our toxicity assays confirmed that C16:0 supplementation partially restored toxicity, suggesting a minor but measurable role for CerS5-mediated 1-deoxyDHCer formation. Similarly, the mitochondrial swelling assays demonstrated that also shorter acyl chain 1-deoxyDHCer species (e.g., m18:0/16:0) do exert a certain toxicity on mitochondria, although to a much lower extent than longer-chain species.

Nonetheless, to preserve the unbiased nature of the screen, we focused on genes meeting the predefined statistical thresholds (p-value and logFC), which best represent robust modifiers of 1-deoxySL toxicity in this setting. We have clarified this point in the revised figure legend and results section.

Fig. 2b,d: When analyzing the toxicity of deoxySphingolipids: If the knockdown of a gene conferred a significant resistance to 1-deoxySa induced toxicity should this gene really be termed “resistance gene” and its pathway “resistance pathway”? Along this, is FADS3 a “sensitizing gene” or a “de-sensitizing gene”? This nomenclature turns the cellular effect upside down.

We agree that the previously used terminology may have been misleading. The designations “resistance gene” and “resistance pathway” were based on an earlier nomenclature convention but may invert the cellular effect and cause confusion. To improve clarity and accuracy, we revised the figure and corresponding text to reflect the functional outcomes more appropriately. Genes whose knockdown mitigates 1-deoxySa-induced toxicity are now described in terms of their sensitizing effects. This adjustment should better align the nomenclature with the biological interpretation for readers of this manuscript.

If claiming that Fig.3d,h displays a “significant reduction” or “significant protection” please provide statistics. Update legend to Fig.3 or provide IC50 values.

This is an important point. In response, we have now included the IC50 values in the revised Figure 3 to quantitatively support the claim of “significant protection.” These values were derived from nonlinear regression of the dose–response curves using a four-parameter logistic model.

We opted to use IC50 (half-maximal inhibitory concentration) as it offers a robust and integrated measure of cytotoxicity across the entire concentration range. IC50 represents the concentration of a compound (in this case, 1-deoxySA) at which a 50% reduction in cell viability is observed. A rightward shift in IC50 in the presence of a protective condition (e.g., genetic knockout or chemical inhibitor) reflects increased resistance to the cytotoxic effect and thus provides a quantitative and statistically meaningful readout of protection. This approach avoids overemphasis on single-point comparisons, which can be misleading due to variability at individual concentrations.

IC50 estimation is a well-established method in pharmacology and toxicology for comparing dose-dependent effects between conditions and is widely used for evaluating the potency of both toxic agents and protective interventions. We now explicitly report these values and adjusted the figure legend accordingly.

If claiming that Fig.4c demonstrates “significant protection” please provide statistics

Thank you for this observation. In the revised Figure 4c and its legend, we now provide IC50 values to support the claim of “significant protection.” In the control condition, 1-deoxySa induced a clear dose-dependent reduction in cell viability, allowing accurate IC50 determination. In contrast, in presence of the ELOVL1 inhibitor, cell viability remained above 50% throughout the tested concentration range, and an IC50 could not be calculated. This marked difference in dose–response profiles provides strong evidence of protection.

IC50 is a widely accepted metric in toxicology and pharmacology, offering an integrative and quantitative assessment of treatment effects. The inability to define an IC50 in the treated condition underscores the robust protective effect of ELOVL1 inhibition.

Additionally, we revised the main text to clarify the interpretation and to reflect these pharmacological findings explicitly.

The statement “These results demonstrate that (22) is metabolically active in the nervous system...” should be used with caution (here and later in the ‘Discussion’). Although the EFFECT is detected in the nervous system, the changed deoxy-SL profile here may result from the inhibitors’ metabolic action outside the nervous system and e.g. lipid trafficking between organs. Has it been shown that the avian blood-brain-barrier is fully established at this developmental stage?

We thank the reviewer for this important and well-considered comment. We have revised the relevant statements in both the Results and Discussion sections to more cautiously reflect the experimental findings.

It is well established that the avian blood–brain barrier (BBB) is not yet fully developed at the embryonic stages corresponding to our treatment (E4) and observation (E7), as supported by prior studies (Risau et al., 1986; Ribatti et al., 1993). Nonetheless, in our study, both d3-1-deoxySA and compound (22) were administered directly onto the embryo. As a result, the presence or maturity of the BBB is not directly limiting for compound accessibility.

Importantly, Extended Figure 4 focuses on the peripheral nervous system, specifically the dorsal root ganglia (DRG), which are not protected by the BBB. The data demonstrate that ELOVL1 inhibitor (22) effectively suppresses the formation of the neurotoxic species 1-deoxyDHCeramide (m18:0/24:1) in the very cell types most vulnerable to 1-deoxySL-induced toxicity—namely, sensory neurons.

This metabolic inhibition is further substantiated by the observed functional rescue of neurotoxicity in the same system. Thus, our data strongly support the notion that ELOVL1i exerts its protective effect at the site of 1-deoxySL-mediated damage, even if secondary contributions from systemic lipid trafficking cannot be excluded.

Omit “Comparable neuroprotective effects were observed in differentiated SH-SY5Y neuroblastoma cells, where (22) prevented neurite collapse following 1-deoxySa exposure (data not shown).” or show the data.

We removed this sentence and all references to the experiment from the manuscript.

From the sentence “Lipidomics profiling confirmed the selective increase in the corresponding LC or VLC 1-deoxyDHCer species following supplementation (Figure 5b)” either “selective” or “LC” should be omitted.

We appreciate the reviewer’s suggestion. The intention of the sentence was to highlight that our assay allows for the selective enrichment of specific acyl chain lengths in 1-deoxyDHCer species, depending on the supplemented fatty acid. For clarity, we have rephrased the sentence in the manuscript to avoid redundancy and ensure accurate interpretation.

Fig. 5b,c and also in the discussion: The authors state that (22) was added in these experiments to avoid further metabolism of the added FAs by ELOVL1. Yet, Fig.3a states that ELOVL1 elongates 20:1, 22:1 and 22:0, which were not added/assayed. A reader might not understand at once why (22) was added at all. In the rebuttal the authors explain this properly but have not added such explanation to the text. This should be corrected – at least with a few sentences and reference(s), if no additional figure is possible/needed.

As shown in Figure 5b, supplemented palmitic acid (16:0) undergoes elongation to stearic acid (18:0), which serves as a precursor for further elongation steps yielding 24:0 and 24:1—both of which are substrates of CerS2 and are directly implicated in the formation of toxic VLC 1-deoxyDHCer species, as illustrated in Figure 3a.

While ELOVL1 preferentially elongates 20:1, 22:0, and 22:1, as described in the study by Ohno et al. (PNAS, 2010, <https://doi.org/10.1073/pnas.1005572107>), our lipidomic profiling demonstrates that elongation from 16:0 to 24:0, 24:1 species also occurs under our experimental conditions. Importantly, ELOVL1 knockdown (Figure 3c,d) and pharmacological ELOVL1 inhibition (Figure 4b,d) substantially reduced levels of the toxic VLC species (e.g., m18:0/24:0 and m18:0/24:1), while shorter species such as m18:0/22:0 remained unchanged or even increased (e.g., m18:0/20:0), despite strong protection from 1-deoxySa-induced toxicity. This data strongly suggest that m18:0/22:0 or m18:0/20:0 are not the mediators of the toxic effects supporting the overall conclusions of the manuscript.

We agree that the rationale for using ELOVL1 inhibitor (22) in the resupplementation assays was not sufficiently explained in the manuscript. We have now revised the Results and Discussion sections to clarify that (22) was added to prevent further elongation of supplemented fatty acids (such as 16:0), which would otherwise result in mixed VLC-1-deoxyDHCer species and confound the interpretation of structure–toxicity relationships.

The legend to Fig. 5b indicates a “vehicle control – BSA alone”. If this is the green bars, please label as such. If not, please clarify.

We thank the reviewer for pointing this out. We agree that the labelling of the vehicle control could have been clearer. In these experiments, the vehicle control consisted of BSA alone, without any supplemented fatty acids. This has now been explicitly clarified in the figure legend. For both the lipidomics (Fig. 5b) and toxicity assays (Fig. 5c), we have revised the figure legends to ensure it is clearly stated that “vehicle control” refers to fatty acid–free BSA complexes.

If claiming that Fig.5d shows “most significant swelling response” please provide statistics. How does the bar look for the positive control (50uM CA2+)?

Statistical significance has now been added to Fig. 5d to support the claim regarding the extent of mitochondrial swelling. The Ca²⁺ (50 μM) positive control has also been included to provide a direct reference for assay responsiveness.

Additionally, we have clarified that the 1-deoxyCer used in this experiment corresponds to a non-native but commercially available 4–5E isomers, as the physiologically relevant 14–15Z isomer is not commercially available as a Ceramide. This limitation has been added to the figure panel, legend, and corresponding section in the main text to avoid confusion and ensure transparency in the interpretation of the results.

In the rebuttal, the authors state: “Our data showed that 1-deoxyDHCer (m18:0/24:1) induces mitochondrial swelling which is rescued by ... BAX inhibition...”. Has this experiment been performed?

We thank the reviewer for this important observation. In our study, we first identified 1-deoxyDHCer species containing 24:0 and 24:1 fatty acyl chains as the principal mediators of 1-deoxySL-induced cytotoxicity. We then demonstrated that 1-deoxyDHCer (m18:0/24:1) induces mitochondrial swelling in isolated mitochondria, consistent with mPTP opening. This was further supported in cellular assays, where treatment with Cyclosporin A (CsA), an mPTP inhibitor, significantly attenuated the toxic effects of 1-deoxySa.

Given the well-established role of BAX in facilitating mPTP-mediated cell death, we extended our investigation using a specific BAX inhibitor. This intervention also rescued cells from 1-deoxySL-induced toxicity, supporting the hypothesis that BAX contributes to the downstream mitochondrial dysfunction.

However, we agree with the reviewer that our previous wording implied a direct causal link between m18:0/24:1-induced mitochondrial swelling and BAX activation, which was not directly tested in isolated mitochondria. Therefore, we have revised the wording in the abstract and discussion to clarify that our findings only suggest a mechanistic link but do not establish direct causality.

The second sentence of the 'Discussion' puts forward a strong message by narrowing it down to a single species. This narrow statement is also repeated in the 'Abstract' (see my comment above). However, the authors show in their swelling assay that m18:0/24:0 shows ~60% and even m18:0/16:0 displays ~50% of the maximum toxicity. If this narrow statement is to be kept at both instances, it would need further support, e.g. analogous experiments as shown in Fig. 5e using m18:0/24:0, m18:0/16:0 and also analogous experiments as shown in Fig.5f using 1-deoxySA + (22) +/- CSA and/or BAXi.

We thank the reviewer for this valuable comment. We agree that the original formulation in the abstract and discussion was overly narrow in attributing the observed toxicity solely to a single species. While our data support that 1-deoxyDHCer (m18:0/24:1) exhibits the highest toxicity, other species such as m18:0/24:0 and m18:0/16:0 also demonstrate toxic effects to a lower extent. Although the mitochondrial swelling assay supports our cell-based findings, it represents a simplified, reductionist model that serves to dissect downstream effects in an isolated and highly controlled setting. As such, it cannot fully replicate the complexity of intracellular lipid flux and metabolism.

To address this, we have revised the abstract and discussion to tone down the emphasis on a single species and to reflect a broader contribution of very-long-chain 1-deoxyDHCer species to toxicity.

We appreciate the reviewer's suggestion to expand these assays to other species and inhibitors and their combinations. While these additional experiments are indeed scientifically interesting and may be explored in future studies, we believe the current dataset sufficiently demonstrates that very-long-chain 1-deoxyDHCer species, particularly those derived from nervonic and lignoceric acid, are the key mediators of 1-deoxySL induced toxicity. Additionally, incorporating multiple lipid species and combined treatments ELOVL1 inhibitors with CsA and BAXi into these assays introduces significant interpretational complexity. Experimentally, such combinations require extensive method development and optimization to ensure data are biologically meaningful and accurately attributable to individual lipid species, which is beyond the main focus of current manuscript.

The revised text better reflects this interpretation of the data and reinforces the main message of the manuscript.

The authors removed some data during the revision. Hence, “mitochondrial fragmentation” should be removed from the ‘Discussion’, too.

We thank the reviewer for this observation. While our intention was to reference previously published findings by Alecu et al., J Lipid Res 2017., we agree that the current version of the manuscript no longer includes supporting data for mitochondrial fragmentation. Accordingly, we have removed the respective statement from the discussion and rephrased the text to ensure consistency with the presented data.

Please provide the source for the individual 1-deoxyDHCer, 1-deoxyCer, or canonical dihydroceramide (DHCer) species used in Fig. 5d.

We thank the reviewer for this comment. The sources and ordering details of all lipid standards used in Fig. 5d—including individual 1-deoxyDHCer, 1-deoxyCer, and canonical DHCer species—have now been added to the Methods section for full transparency and reproducibility.

All abbreviations should be harmonized e.g. Ceramide synthase is abbreviated as CerS and CERS.

Text was corrected.

The use of "knockout" and "knockdown" at several instances in the text needs to be doublechecked.

Text was double checked.

In the ‘Author contributions’ some initials remain flipped, e.g. MA <> AM or EY <> YE ...

Initials in the author contributions were corrected.

REVIEWERS' COMMENTS

Reviewer #2 (Remarks to the Author):

The authors have made corrections and clarifications to some of my previous comments. However, some points remained to be addressed:

1. The information and citation of the CRISPRi v2 library used in this study is missing. Is it a commercial library or a custom library? Please provide the list of targeted genes and the corresponding sgRNA sequences included in the library.

The list of target genes and sgRNAs is provided as **Source Data 2**

2. The screen was CRISPRi-based, which does not cut DNA. However, in the diagram in Figure 2a a scissor icon was used to represent CRISPR cutting, which does not accurately reflect the screening platform. Please replace or correct this icon/label to accurately depict CRISPR interference (CRISPRi) rather than nuclease-mediated cutting.

The symbol has been removed

3. The authors stated that the raw data associated with the screen are not available. Since the screen results are a key dataset for the paper, the absence of original/raw data prevents evaluation and reproduction of the findings. If the authors cannot provide the raw data, they need either repeat the screen and supply the raw data, or remove the screen results from the current manuscript.

We have contacted Washington University regarding the recovery of the raw sequencing data. Unfortunately, the raw sequencing data are no longer retrievable. However, as the screen results were validated in several independent assays, we believe that the validity of the screen has been orthogonally confirmed.

4. The processed data are also missing in the current revision. These should include, at minimum: sgRNA count tables, the output files from the ScreenProcessing pipeline, and the dataset used to generate Figure 2b. Please submit these datasets as supplementary tables in the revision.

The intermediate and processed sequencing datasets have been included as **Source Data 3**.